

# Application of the ECT9 protocol for radiocarbon-based source apportionment of carbonaceous aerosols

Lin Huang[1]*, Wendy Zhang[1], Guaciara M. Santos[2], Blanca T. Rodríguez[2], Sandra R. Holden[2], Vincent Vetro[1], Claudia I. Czimczik[2]*

[1]Climate Research Division, Atmospheric Science & Technology Directorate, Environment and Climate Change Canada, Toronto, ON M3H 5T4, Canada
[2]Department of Earth System Science, University of California, Irvine, CA 92697-3100, USA

*Correspondence to: Lin Huang (lin.huang@canada.ca); Claudia I. Czimczik (czimczik@uci.edu)

Key words: Radiocarbon, organic carbon, elemental carbon, black carbon, Arctic, EnCan-total-900, SRM1649a

**Abstract:** Carbonaceous aerosol is mainly composed of organic carbon (OC) and elemental carbon (EC). Both OC and EC originate from a variety of emission sources. Radiocarbon ($^{14}$C) analysis can be used to apportion bulk aerosol, OC, and EC into their sources. However, such analyses require the physical separation of OC and EC.

Here, we apply of ECT9 protocol to physically isolate OC and EC for $^{14}$C analysis and evaluate its effectiveness. Several reference materials are selected, including: two pure OC (fossil "adipic acid", contemporary "sucrose"), two pure EC (fossil "regal black" and "C1150"), and three complex materials containing contemporary and/or fossil OC and EC ("rice char" and NIST urban dust standards "SRM1649a", i.e., bulk dust and "SRM8785", i.e., fine fraction of re-suspended SRM1649a on filter). The pure materials were measured for their OC, EC and total carbon (TC) mass fractions and corresponding carbon isotopes to evaluate the uncertainty of the procedure. The average accuracy of TC mass, determined via volumetric injection of a sucrose solution, was approximately 5%. Ratios of EC/TC and OC/TC were highly reproducible, with analytical precisions better than 2% for all reference materials, ranging in size from 20 to 100 µg C. Consensus values were reached for all pure reference materials for both $\delta^{13}$C and FM$^{14}$C with an uncertainty of <0.3‰ and approximately 5%, respectively. The procedure introduced 1.3±0.6 µg of extraneous carbon, an amount compatible to that of the Swiss_4S protocol.

In addition, OC and EC were isolated from mixtures of pure contemporary OC (sucrose) with pure fossil EC (regal black) and fossil OC (adipic acid) with contemporary EC (rice char EC) to evaluate the effectiveness of OC and EC separation. Consensus FM$^{14}$C values were reached for all OC (~ 5-30 µg) and EC (~10-60 µg) fractions with an uncertainty of <5%. We found that the ECT9 protocol efficiently isolates OC or EC from complex mixtures. Based on $\delta^{13}$C measurements, the average contribution of charred OC to EC is likely less than 3% when the OC loading amount is less than 30 µg C.

Charring was further assessed by evaluating thermograms of various materials, including aerosol samples collected in the Arctic and from tailpipes of gasoline or diesel engines. These data demonstrate that the ECT9 method effectively removes pyrolyzed OC. Thus, the ECT9 protocol, initially developed for concentration and stable isotope measurements of OC and EC, is suitable for $^{14}$C-based apportionment studies for environment samples, including µg C-sized samples from Arctic environments.



## 1 Introduction

Carbonaceous aerosol is a major component (15-90%) of airborne particulate matter (PM) (Jimenez et al., 2009; Putaud et al., 2010; Yang et al., 2011; Hand et al., 2013; Ridley et al., 2017), and a complex mixture composed of mainly light-scattering organic carbon (OC) and highly-refractory, light-absorbing elemental carbon (EC, also referred to as black carbon) (Pöschl, 2005). The OC and EC fractions play important and often distinct roles in climate (Bond et al., 2013; Hallquist et al., 2009; Kanakidou et al., 2005; Laskin et al., 2015), air pollution and human health (Cohen et al., 2017; Grahame et al., 2014;

Janssen et al., 2012). Moreover, both OC and EC were identified as short lived climate forcers (SLCFs) by the IPCC expert meeting (https://www.ipcc-nggip.iges.or.jp/public/mtdocs/1805_Geneva.html) in 2018. To develop and monitor the efficiency of mitigation strategies for both climate change and air pollution, it is required to have a better understanding of the temporal and spatial dynamics of OC and EC emission sources.

The majority (>50%) of carbonaceous aerosol is OC, which has a wide size range. Coarse OC (in $PM_{10}$) consists of plant

debris, microorganisms, fungal spores, and pollen. Fine OC (in $PM_{2.5}$) is formed predominantly via the oxidation or nucleation/coagulation of volatile organic compounds, such as mono- and sesquiterpenes, from both biogenic and anthropogenic sources (Shrivastava et al., 2017), but can also be directly emitted from combustion sources (Hallquist et al., 2009; Fuzzi et al., 2015; Liggio et al., 2016). In contrast, EC is found primarily in fine particles, e.g., $PM_{1.0}$ or smaller (Chan et al., 2013; Bond et al., 2013). It is emitted through incomplete combustion of fossil fuels and biomass/biofuels (Bond et al.,

2013; Huang et al., 2010; Evangeliou et al., 2016; Winnie et al., 2016; 2017; 2019).

Measuring the isotopic signature and composition, i.e. radiocarbon ($^{14}C$) content and stable isotope ratio ($^{13}C/^{12}C$) of aerosol, offers a powerful tool for quantifying the sources of bulk aerosol and its OC and EC fractions. Aerosol $^{14}C$ content can be used to quantify the relative contributions from contemporary biomass and fossil sources (Heal, 2014). $^{14}C$ is a naturally occurring radioisotope (5,730-year half-life) produced in the atmosphere. After its oxidation to carbon dioxide ($^{14}CO_2$), $^{14}C$

enters the food chain through photosynthesis so that all living organisms are labeled with a characteristic $^{14}C/^{12}C$ ratio and described as "modern" carbon. Materials containing carbon older than about 50,000 years ($^{14}C<<^{12}C$) are described as "fossil" carbon. Over the past centuries, the $^{14}C$ content of the atmosphere has undergone distinct changes (Graven, 2015; Levin et al., 2010): Anthropogenic combustion of fossil fuels emit $^{14}C$-depleted carbon into the atmosphere (i.e. dilute the proportion of $^{14}C$ relative to $^{12}C$). In contrast, nuclear weapons testing doubled the $^{14}C$ content of $CO_2$ in the Northern

Hemisphere in the mid-20[th] century, followed by mixing of this bomb-derived $^{14}C$-enriched carbon into the ocean and biosphere. Similarly, aerosol stable isotope ratios provide insight to different types of anthropogenic sources (e.g. combustion of solid and liquid vs. gaseous fossil fuels). However, $^{13}C$ data cannot distinguish emissions from mixed fossil fuel combustion and live C3 plant biomass (Huang et al., 2006; Winiger et al., 2016). Thus, isotope-based source apportionment studies become particularly insightful when both $^{14}C$ and stable carbon isotopes are considered (Andersson et al., 2015;

Winiger et al., 2016, 2017) or when combined with analyses of specific source tracers, such as levoglucosan or potassium for wood burning emissions (Szidat et al., 2006; Zhang et al., 2008) and/or remote sensing data and modeling analysis (Barrett et al., 2015; Mouteva et al., 2015b; Wiggins et al., 2018).



The objective of this study is to evaluate the effectiveness of separating OC and EC via the ECT9 (EnCan-Total-900) protocol (Huang et al., 2006; Chan et al. 2010; Chan et al., 2019) for $^{14}$C-based source apportionment studies of carbonaceous

aerosols. The ECT9 technique was originally developed to physically separate OC and EC mass fractions for concentration quantification and stable carbon isotope analysis. This protocol has been used since 2006 to monitor carbonaceous aerosol mass concentrations and stable isotope composition over Canada, including in the Arctic at Alert, as part of the Canadian Aerosol Baseline Measurements (CABM) Network by Environment & Climate Change Canada (Chan et al., 2010; 2019; Eckhardt et al., 2015; Sharma et al., 2017; Xu et al., 2017; Leaitch et al., 2017; 2018; Huang, 2018). It has also been used to

monitor carbonaceous aerosol over China (Yang et al., 2011). Furthermore, EC concentration measurements made with the ECT9 protocol correlate well with those derived from light absorption by an aethalometer as well as refractory black carbon (rBC) using a Single Particle Soot Photometer (SP2) (Sharma et al., 2017; Chan et al., 2019). It was demonstrated that the ECT9 protocol can be used to quantify OC/EC concentrations and provide source information at the same time.

The ECT9 protocol is a thermal evolution analysis (TEA) protocol which is different from commonly used thermal optical

analysis (TOA) methods for monitoring air quality, such as the Interagency Monitoring of Protected Visual Environments (IMPROVE) protocol (Chow et al., 2001; Watson et al., 2007), the National Institute for Occupational Safety and Health protocol (NIOSH method 5040, Birch, 2002), as well as the European Supersites for Atmospheric Aerosol Research (EUSAAR) protocol (Cavalli et al., 2010). In those protocols, the OC fraction is thermally desorbed from filter samples in an inert helium (He) atmosphere at relatively lower temperatures and the EC fraction is combusted at higher temperatures by

introducing oxygen ($O_2$) in He stream while the filter reflectance or transmittance for a laser signal is continuously monitored. During the analysis, a fraction of the OC may char (forming pyrolyzed OC or POC), causing the transmittance or reflectance to decrease. While TOA methods use the changes in laser signal to mathematically correct for POC within the measured EC fraction, the ECT9 protocol aims to minimize or remove POC, together with carbonate carbon (CC), during an intermediate temperature step of 870°C in pure He via high temperature evaporation (Chan et al., 2019). With much longer

retention times at each temperature step (see Methods) and without either reflectance or transmittance used, the ECT9 protocol effectively isolates OC, POC+CC, and EC.

It should be noted that other methods have been also developed mainly for $^{14}$C analysis of OC and EC, such as the CTO-375 (Zencak et al., 2007), the Swiss_4S protocol (Mouteva et al., 2015a; Zhang et al., 2012), or hydropyrolysis (Meredith et al., 2012; Zhang et al., 2019), which use distinct temperature protocols, gas mixture and/or remove water-soluble OC or

inorganic carbon prior to EC analysis. In contrast to the ECT9 protocol, however, these approaches differ substantially from the protocols that are widely used for monitoring OC/EC mass concentrations in the field, which limits the relevance of this data for improving the representation of carbonaceous aerosols in chemical transport models.

Here we analyzed the $^{14}$C content of OC and EC fractions (<100 µg C) isolated with the ECT9 protocol from four pure fossil and contemporary reference materials. These materials were analyzed on their own to quantify the amount and source

(modern or fossil) of extraneous carbon introduced by the procedure as well as its reproducibility. Mixtures of two reference materials were measured to elucidate how efficiently the ECT9 protocol isolates OC from EC. In addition, we investigated the laser signals of three reference materials and three aerosol samples (tailpipe emissions, ambient aerosol from Alert, and SRM8785) to assess how efficiently the ECT9 protocol removes POC. Our evaluation of the ECT9 protocol on its ability to



physically separate OC from EC for [14]C-based source apportionment studies significantly expands the existing opportunities

for characterizing and monitoring sources of carbonaceous aerosol at regional or global scales at the same time providing

solid base for EC and OC concentration measurements.

## 2 Methods

### 2.1 The ECT9 protocol for the physical separation of OC and EC

The ECT9 protocol was developed at the carbonaceous aerosol & isotope research (CAIR) lab of Environment and Climate

Change Canada (ECCC) to quantify the amount of OC and EC in carbonaceous aerosol and their $\delta^{13}$C values (Huang et al.,

2006; Chan et al., 2010; 2019). Carbon fractions are isolated with an OC/EC analyzer (Sunset Laboratory Inc.) coupled to a

custom-made gas handling and cryogenic trapping system for $CO_2$ collection from OC and EC fractions (Fig. 1a). The

fractions are separated based on their thermal refractory. Specifically, carbon fractions are released by the ECT9 protocol in

three steps (Fig. 1b): (1) OC at 550°C for 600 seconds in pure He (99.9999% purity); (2) POC and CC at 870°C for 600

seconds in pure He; and (3) EC at 900°C for 420 seconds in a mixture of 2% $O_2$ with 98% He. All fractions are fully oxidized

to $CO_2$ by passing through a furnace containing $MnO_2$ maintained at 870°C. For concentration determination, the $CO_2$ passes

through a methanator at 500°C, is converted to $CH_4$, and quantified with a flame ionization detector. For isotope analysis, the

$CO_2$ is cryo-trapped with liquid $N_2$ (-196°C) in a U shaped glass trap, purified on a vacuum system (to remove He), sealed

into a Pyrex ampoule, and analyzed for its $\delta^{13}$C ratio with an Isotopic Ratio Mass Spectrometer (IRMS), i.e., MAT253 or

FM[14]C with an Accelerated Mass Spectrometer (AMS).

### 2.2 Reference materials and their composition

To evaluate the ECT9 method for separating OC and EC for [14]C analysis, we isolated and measured the [13]C and [14]C content

of the OC or EC fraction or TC from 5-6 modern or fossil reference materials (Table 1), including two pure OC (adipic acid,

sucrose), two EC (C1150, regal black), and two natural OC/EC-mixtures (rice char and urban dust SRM1649a).

Some of the reference materials have previously been utilized to compare different protocols that quantify OC/EC fractions

(Hammes et al., 2007; Willis et al., 2016) as well as determine the mass of extraneous carbon introduced during OC/EC

isolation from carbonaceous aerosol (Mouteva et al., 2015a). Table 1 provides an overview of their chemical compositions,

i.e., total carbon contents and relative fraction of OC and EC, respectively (for individual measurements see Table S1).

Primary methods (i.e., gravimetric or volumetric) are used for mass loading of the materials, whereas the mass of TC, OC,

and EC are quantified via the ECT9 thermal protocol. Based on repeat injections of sucrose results (20-80 μg sucrose, n

=117), the accuracy of the TC mass is about 5%. The reproducibilities of both OC/TC and EC/TC percentages are 2% or

better. Although uncertainties of weighing pure EC mass (i.e., Regal black and C1150) via microbalances are relatively large

(due to static electricity and variable relative humidity), the EC/TC and OC/TC ratios for all reference materials are highly

reproducible (one s.d. <2%). The results show that the two EC materials (i.e., regal black and C1150) contain 97% and 98%

EC, with only 3% and 2% OC, respectively. The two OC materials (i.e., sucrose and adipic acid) are 99% and 100% OC, and



less than 1% EC (likely due to charred OC contribution), respectively. Thus, the materials are suitable for the purpose of this study.

We also analyzed the $^{13}$C and $^{14}$C isotopic composition of each reference material, using off-line combustions and ECT9 coupled with cryo-purification to convert them into $CO_2$. The results are summarized in Table 2 (for individual results see

Tables S2 & S3). The $^{14}$C analysis of µg C-sized carbonaceous aerosol samples requires the assessment of extraneous carbon (Santos et al., 2010). This is achieved by measuring multiple smaller-sized materials with known $^{14}$C content. Consequently, the results in Table 2 are critical, as those $^{14}$C values provide the reference for quantifying the extraneous carbon introduced during the isotope analysis procedures.

### 2.3 Isolation of OC, EC or TC with the ECT9 protocol and purification of $CO_2$

The isotopic analysis of carbonaceous aerosol via the ECT9 system involves three steps (Fig. 1a): 1) OC and EC isolation/$CO_2$ collection and 2) $CO_2$ purification, followed by 3) isotope analysis for either $^{13}$C/$^{12}$C by IRMS or $^{14}$C by AMS (coupled measurements of $^{13}$C/$^{12}$C and $^{14}$C/$^{12}$C of µg C- sized graphite targets), as desired.

The initial masses of the pure reference materials ranged from 5 to 47 µg C (n=3-13; Table S6), whereas those for the mixed materials ranged from 5-30 µg C for OC and 5-60 µg C for EC (n=5-6; Table S7). The loaded mass of each material was

determined via a microbalance (MX5, Mettler Toledo or CCE6, Sartorius) with the lowest reading to 1 µg C or 0.1 µg C, respectively. OC materials were dissolved in MQ-water with known volumes and volumetrically loaded onto a pre-cleaned quartz filter punch (1.5 cm$^2$, Pall Canada Limited). EC (i.e., Regal black and C1150) and mixed materials (rice char or SRM 1649a), which cannot be completely dissolved in water, were directly weighed onto pre-cleaned quartz filter punches. These filters were pre-combusted at 900°C in a muffle furnace overnight and wrapped into aluminum foil before cooling below

200°C. A filter punch with the loaded mass was put into the Sunset analyzer and analyzed with the ECT9 protocol. OC and EC were separated and the combusted OC or EC fractions as $CO_2$ were cryo-collected in a U-shaped flask submerged in liquid $N_2$ (Fig. 1a, step 1). Then, this flask containing $CO_2$ and He was connected to a vacuum line with 4 cryo-traps and several open ports (Fig. 1a, step 2), where the $CO_2$ is purified by sequential distillation when passing cryo-traps 1 through 3. Finally the pure $CO_2$ is transferred and sealed into a 6 mm glass ampoule for $^{13}$C or $^{14}$C analysis.  Pressure is read by a Pirani

gauge before sealing the ampoule for an estimation of the amount of gas, and consequently, sample size determination as µg C.

### 2.4 $^{14}$C measurements

At the KCCAMS facility, the OC and EC fractions or TC (in form of $CO_2$) were reduced to graphite on iron powder via hydrogen ($H_2$) reduction using equipment and protocols specifically developed for smaller-sized (≤15 µg C) samples (Santos

et al., 2007b; 2007a). Briefly, sample-$CO_2$ was introduced into a vacuum line, cryogenically isolated from any water vapor, monometrically quantified, and then transferred to a heated reaction chamber, where it was mixed with $H_2$ and reduced to filamentous graphite. To characterize the graphitization, handling and AMS analysis, two relevant standards (Oxalic Acid II as modern carbon and Adipic acid as fossil carbon), with similar size ranges of the samples prepared via ECT9, were also



processed into graphite. The graphite was then pressed into aluminum holders and loaded into the AMS unit alongside

measurement standards (Table S6) and blanks for $^{14}$C measurement (Beverly et al., 2010). The data are reported in fraction
modern carbon (FM$^{14}$C), following the conventions established by Stuiver and Polach (1977) and also described elsewhere
(Trumbore et al., 2016).

To establish consensus values (Table 2), we also analyzed the $^{14}$C content of the bulk reference materials ranging in size from
0.06 to 1 mg C, using our standard combustion and graphitization methods. Larger aliquots of material were weighed into

pre-combusted quartz tube with 80 mg CuO, evacuated, and combusted at 900°C for 3 hours. The resulting $CO_2$ was
cryogenically purified on a vacuum line, reduced to graphite using a closed-tube zinc-reduction method (Xu et al., 2007), and
analyzed as described above.

**2.5 Quantification of extraneous carbon**

Any type of sample processing and analysis introduces extraneous carbon ($C_{ex}$). Therefore, the measured mass of any sample

will include the mass of this sample and of any $C_{ex}$ incorporated throughout the analysis [Eq. 1]:

$$m_{spl\_meas} = m_{spl} + m_{ex}$$  [Eq. 1],

where $m_{spl\_meas}$, $m_{spl}$, and $m_{ex}$ are the measured and theoretical mass of the sample and of $C_{ex}$, respectively. For small
samples (with a mass of a few µg C), the mass of $C_{ex}$ can compete with or overwhelm the sample mass and cause the
measured FM$^{14}$C value of a sample to deviate from its consensus value.

Here, we estimated the mass of $C_{ex}$ introduced during the ECT9 protocol and the $^{14}$C analysis following Santos et al. (2010),
where $C_{ex}$ is understood to consist of a modern and of fossil component [Eq. 2]:

$$m_{ex} = m_{mex} + m_{fex}$$  [Eq. 2],

where $m_{mex}$ and $m_{fex}$ is the mass of the modern and fossil $C_{ex}$, respectively.

Following an isotope mass balance approach, the measured isotopic ratio ($^{14}$C/$^{12}$C) of a sample ($R_{spl\_meas}$) can be expressed

as [Eq. 3].

$$R_{spl\_meas} = \frac{m_{spl}R_{spl}+m_{mex}R_m+m_{fex}R_f}{m_{spl\_meas}}$$  [Eq. 3],

where $R_{spl}$ is the theoretical isotopic ratio of the sample, and $R_m$ and $R_f$ are the consensus isotopic ratios of a
modern and fossil standard, respectively. This equation can be further simplified because $R_f$ is 0. $R_m$ is determined
by measuring regular-sized aliquots of this reference material. In addition, all $^{14}$C/$^{12}$C ratios are corrected for isotope

fractionation using their δ$^{13}$C measured alongside $^{14}$C on the AMS (Beverly et al., 2010).

The mass of modern $C_{ex}$ can be quantified by analyzing fossil reference materials, which are highly sensitive to
modern and insensitive to fossil pollutants. Based on [Eq. 3] the measured isotopic ratio of the fossil reference
($R_{f\_meas}$) can be expressed as [Eq. 4]:





$$R_{f\_meas} = \frac{m_{mex}R_m}{m_{spl\_meas}}$$ [Eq. 4]

The smaller the mass of the fossil reference material, the greater the effect of the constant mass of modern $C_{ex}$ on the isotope ratio of the fossil reference material, i.e. $R_{f\_meas}$ deviates toward $R_m$.

Similarly, the mass of fossil $C_{ex}$ can be quantified by analyzing modern reference materials. With decreasing mass, the measured isotopic ratio of the modern reference ($R_{m\_meas}$) will deviate more strongly from $R_m$ (toward $R_f$). Based on [Eq. 1-3] and assuming $m_{spl} \gg m_{mex}$, the $R_{m\_meas}$ can be expressed as [Eq. 5]:

$$R_{m\_meas} = \frac{m_{spl}R_m + m_{mex}R_m}{m_{spl\_meas}} \approx \frac{(m_{sp\ meas} - m_{fex})R_m}{m_{spl\_meas}}$$ [Eq. 5]

Finally, we can calculate the $C_{ex}$-corrected isotope ratio of an unknown sample ($FM_{spl\_cor}$). This value reported as the ratio between the theoretical isotopic ratio of this sample and the accepted value of a modern standard $\left(R/R_m\right)$ also known as Fraction Modern (FM; with all $R$ corrected for stable isotope fractionation). This data is reported as [Eq. 6]:

$$FM_{spl\_cor} = \frac{R_{spl}}{R_m} \approx \frac{R_{spl\_meas} - R_{f\_meas}}{R_{m\_meas} - R_{f\_meas}} \approx FM_{m*} \times \frac{\left[\frac{R_{spl\_meas}}{R_m} - \frac{m_{mex}}{m_{spl\_meas}}\right]}{\left[1 - \frac{m_{mex}}{m_{spl\_meas}} - \frac{m_{fex}}{m_{spl\_meas}}\right]}$$ [Eq. 6],

where $FM_{m*}$ is determined from the direct measurement of the modern primary reference material (OX1) used to produce six time-bracketed graphite targets measured in a single batch, after isotopic fractionation correction and normalization (Santos et al., 2007a,b). The individual uncertainty of $FM_{spl\_cor}$ is determined from counting statitics and by propagating the quantified blanks using a mass balance approach. Long-term and continuous measurements

of various types of blanks indicate that the mass of $C_{ex}$ within one analytical method or system can vary as much as 50% (see Santos et al., 2010; Fig. 1). Therefore, we applied a 50% error in $m_{fex}$ and $m_{mex}$ from long-term measurements of variance in $m_{ex}$ of small samples (Santos et al., 2007a).

In this study, we used a multi-step approach to quantify $m_{ex}$ introduced by the ECT9 protocol and [14]C analysis. First, we quantified $m_{ex}$ introduced during [14]C analysis by analyzing different masses of our bulk reference

materials. Extraneous carbon is introduced during sealed tube combustion and graphitization followed by graphite target handling and measurement. Typically, [14]C analysis contributes a small portion to $m_{ex}$ (Mouteva et al., 2015a; Santos et al., 2010). Second, we quantified the portion of $m_{ex}$ added during the isolation of OC and EC with the ECT9 protocol. This portion of $m_{ex}$ allows us to determine the practical minimum sample size limit for the entire method, including $m_{ex}$ contributions from filter handling before OC/EC analysis, instrument separation, and transfer

to cryo-collection system and Pyrex ampoules. To isolate this portion, we quantified $m_{ex}$ of the entire procedure (ECT9 protocol plus [14]C analysis) by analysing the [14]C signature of OC and EC from different masses of a large set of reference materials, and then subtracted the portion of $m_{ex}$ introduced during [14]C analysis.





## 3. Results and Discussion

### 3.1. Recovery estimation

The reference materials used in this study, including the modern and fossil endmembers (i.e., the major carbon sources) found in carbonaceous aerosol, and their TC, OC, and EC concentrations are shown in Table 1. Reference materials were separated into OC, EC, or TC using the ECT9 method at ECCC's CAIR lab (Fig. 1) and analyzed for their $^{14}$C content at UC Irvine's KCCAMS facility.

Fig. 2 shows the cross-validation of carbon-mass between the mass determined at ECCC's CAIR lab and the mass
quantified at UC Irvine's KCCAMS lab indicating a very good positive correlation ($R^2$ = 0.93 for pure materials and $R^2$ = 0.95 for two-material-mixtures in Fig. 2a and 2b, respectively). It is suggested that no major losses or gains of carbon occurred during the entire analytical process and the overall recovery was close to 100%, with a 5% uncertainty for samples ranging in size from about 5 to 60 µg C.

### 3.2 Quantification of extraneous carbon and its sources

All types of samples, regardless of size, show deviations in their measured FM$^{14}$C value from their consensus values to certain degree due to $C_{ex}$ introduced during sample analysis. In µg C-sized samples (mass <15 µg C), significant bias from any $C_{ex}$ can be observed, because $C_{ex}$ constitutes a large fraction of the total sample. Previous work (using solvent-free analytical protocols) has shown that modern $C_{ex}$ is introduced mostly through instrumentation and sample handling techniques, while fossil $C_{ex}$ originates from iron oxide used as a catalyst for the reduction of $CO_2$ to graphite prior to AMS
analysis (Santos et al., 2007a; 2007b).

The FM$^{14}$C values of the pure modern or fossil reference materials generally agreed with their accepted FM$^{14}$C values for both OC and EC fractions (within approximately 5% uncertainty, Fig. 3 and Table S6) after applying a constant amount $C_{ex}$ correction in FM$^{14}$C determination. This constant $C_{ex}$ is a critical prerequisite for accurately correcting the FM$^{14}$C value of unknown samples. Hence, our data demonstrated that the ECT9 method (and subsequent $^{14}$C analysis) introduces a small,
reproducible amount of $C_{ex}$.

According to equations [4]-[5] in section 2.5, $C_{ex}$ can be quantified by measuring FM$^{14}$C of pure modern or fossil materials with different sizes. Fig. 3 demonstrates that regardless what $^{14}$C content are in carbon fractions isolated from the reference materials and what sizes they are, the corrected FM$^{14}$C values match with consensus value within propagated uncertainty.

To evaluate the suitability of ECT9 for $^{14}$C analysis of aerosol samples, a comparison is made between the results of a
published method (i.e., Swiss_4S) and those of ECT9. The two protocols are listed in Table 3 and their $C_{ex}$ distribution is shown in Table 4. The total amount of $C_{ex}$ introduced by the complete procedure through ECT9, and determined based on all reference materials, was 1.3±0.6 µg C, with 70% originating from contamination with modern carbon (Table 4). The isolation of OC and EC with the ECT9 protocol introduced 65% of total $C_{ex}$ (0.85 out of 1.35 µg C ), with 65% derived from modern carbon. Overall, the total amount of $C_{ex}$ introduced during OC/EC isolation with the ECT9 protocol is comparable to that for
the Swiss_4S protocol established at UC Irvine within uncertainties (Table 3, Mouteva et al. (2015a)). Thus, it is


demonstrated that the ECT9 protocol serves as a suitable alternative for the $^{14}$C analysis of aerosol samples with masses >2 μg C.

### 3.3 Effectiveness of OC/EC separation

To investigate the effectiveness of the ECT9 to separate OC from EC in more complex mixtures with minimizing OC into the
EC fraction via pyrolysis, mixtures of the modern and fossil reference materials (Table 2) were used for measuring $\delta^{13}$C (Table S4 - S5) and FM$^{14}$C (Table S7).

First, it was found that the FM$^{14}$C values of OC and EC fractions isolated from mixtures of pure sucrose (modern OC) and pure regal black (fossil EC) were within the measurement uncertainty of their accepted FM$^{14}$C values, after correction for a constant amount of $C_{ex}$ (Fig. 4) for samples with 5 – 34 μg OC carbon and 10 – 60 μg EC carbon, showing a good separation
of OC from EC. This amount of $C_{ex}$ was identical to that applied to the pure reference materials above, further corroborating the constant background introduced by the ECT9 protocol and $^{14}$C analysis.

Next, the mixtures of fossil adipic acid (pure OC) and modern rice char (mixture of OC and EC) were isolated and analyzed. It was found that after correction for $C_{ex}$, the FM$^{14}$C values of the OC (from the mixture) were systematically greater than the consensus value of the pure adipic acid, i.e., a FM$^{14}$C of zero (Fig. 5a), indicating that there was certain level of modern
fraction contributed to the measured OC from the modern rice char. This was because rice char contains about 14% of modern OC (Table 1).

To confirm that ECT9 could remove OC contained in rice char, an additional step was taken before mixing modern rice char's EC with the fossil OC (adipic acid). Specifically, we stripped the OC fraction of rice char by running rice char (on a filter) through the ECT9 protocol. Adipic acid (fossil OC) was then injected onto the filter with the remaining rice char-EC.
The results show that the FM$^{14}$C of OC values of this mixture lie well within the expected range of the consensus value (Fig. 5b) after a $C_{ex}$ correction as described above, demonstrating an excellent remove of rice char OC.

In both mixtures (fossil adipic acid with modern bulk rice char or rice char-EC), the corrected FM$^{14}$C values of the isolated EC fractions were within the expected range for the rice char reference material (Fig. 5c, d). This provides further evidence that the ECT9 protocol isolates modern EC from fossil OC with no obvious evidence of transferring fossil OC into the EC
fraction. Together, the three sets of mixing experiments (Figs. 4 & 5) provide strong evidence for the effectiveness of separating OC from EC via ECT9 protocol.

In addition to FM$^{14}$C measurements, $\delta^{13}$C measurements in mixtures of OC and EC can also provide quantitative information on the effectiveness of OC and EC separation via ECT9. Various amounts of sucrose (pure OC, 10 – 30 μg C) were first mixed with varying amounts of Regal black (pure EC, 20 – 66 μg C). The mixtures were then physically separated into OC
and EC fractions by ECT9 for $\delta^{13}$C measurements. The measured $\delta^{13}$C values of OC and EC from these mixing experiments are listed in Table S4. Based on the $\delta^{13}$C values of individual pure reference materials (Table S3) and a two-end-member mixing mass balance, it is estimated that the average fraction contributed into each other in the mixtures (i.e., sucrose fraction into Regal black or vice versa) was likely less than 3% (Table S5).



### 3.4 Charring evaluation & POC removal using the ECT9 protocol

It is known that some of OC (e.g., oxygenated OC or water soluble OC) would char to form pyrolyzed organic carbon (POC) when heated in an inert He atmosphere, darkening the filter (Chow et al., 2004; Watson et al. 2005) and causing decreased laser signals due to light-absorption of charred OC. In most TOA protocols, this POC would combust and contribute to EC when $O_2$ is added. However, POC can be also be gasified and released as CO at high temperatures (>700°C) with limited $O_2$ supply, e.g., oxygenated OC at 870°C (Huang et al., 2006; Chan et al., 2010;

2019). Most TOA protocols estimate POC by quantifying the mass associated with reflectance/transmittance changes, i.e., the mass released between the time when $O_2$ is introduced and the OC/EC split point (where the reflectance/transmittance returns to the initial value). In contrast to other TOA protocols, ECT9 defines POC as the mass released at the temperature step of 870°C (during a period of 600 seconds). This includes charred OC, calcium carbonate ($CaCO_3$) that decomposes at 830°C, and any refractory OC not thermally released at 550°C (Huang et al.,

2006; Chan et al., 2010; 2019).

Although ECT9 do not use laser signals to quantify POC, it is expected that the changes of laser signals during the stage of 870°C would provide useful information about POC. Thus, four sets of samples were selected, including those of pure reference materials and ambient aerosol samples from different sources with heavy or light mass loading (e.g., those Arctic sample filters from different seasons) to evaluate the possible charring via ECT9. Their

thermograms are shown in Figures 6 to 9.

Figure 6 shows thermograms of pure or bulk references for Regal black, sucrose, and rice char, respectively. It is observed in all three that the laser transmittance signals first decrease and then increases again during the 870°C step, and that they return to their initial values just before EC is released at the next step of 900°C. This demonstrates that the ECT9 method minimizes POC-contributions to the EC fraction.

The thermograms of aerosol (on filters) collected directly from tailpipe exhaust of a diesel engine vehicle and a gasoline engine passage car, respectively are shown in Figure 7. These data suggest that the amount of POC generated during analysis are sample/matrix dependent. Specifically, the mass fraction during the 870°C temperature is larger for the gasoline than the diesel engine. This finding supports previous work showing that POC is proportional to the amount of oxygenated OC (Chan et al., 2010). It is noticed that the laser signal reaches the

initial value before the EC step, further demonstrating that the charring contribution to EC is minimized.

Another set of thermograms of two total suspended particle filter samples collected during the summer (August) and winter (December) of 2015 at an Arctic site (i.e., Alert) are shown in Figure 8. More details about these samples can be found in Wex et al. (2019). The laser signal patterns are similar to those shown in Figures 6 & 7, yet more pronounced. During the 550°C step, the laser signals decrease. During the 870°C step, the signals further decrease,

then increase, and finally increase to their initial point before EC is released at 900°C. These thermograms further demonstrate ECT9 is able to minimize POC by gasification.



Finally, the thermographs of NIST urban dust reference material SRM 8785 (the re-suspended SRM 1649a urban dust with a fine fraction <2.5 μm collected on quartz filter) analyzed with ECT9 and Swiss_4S are shown in Figure 9. Both thermograms obtained with the ECT9 method (Fig. 9 a&b) show the similar patterns as those in Figs. 6-8,

i.e., the laser signals reaching the initial value just before the EC release at 900°, suggesting that the charring contribution to EC is minimized during the stage of 870°C even though some POC might remain.

In the thermogram obtained with the Swiss-4S protocol (Fig. 9c), the laser signal increases from the beginning of the run while the first two stages (375°C and 475°C) are under the conditions of pure $O_2$ stream, inferring that light absorbing carbon is released during the first two OC stages. The laser signal continues to increase while the

temperature increases up to 650°C (the third stage) under the pure He gas stream, indicating that no charred OC is formed. However, when the temperature starts decreasing from 650°C, the laser signal decreases, indicating POC formation below that temperature. This signal decrease continues until the beginning of the next pure $O_2$ stage. It is important to note that to obtain EC fraction, the Swiss-4 (Table 3) method calls for filter sample pre-treatment, i.e., extraction with water before the thermal separation of OC/EC to minimize the contribution of charred OC from the

3rd stage to EC at the 4th stage. While it is difficult to make direct comparisons between OC and EC from b) and c) in Figure 9, laser profiles from those thermograms indicate that in both cases charred OC is negligible or minimum.

Together, the thermograms (Figs. 6-9) elucidate that the ECT9 protocol can effectively remove or minimize charred OC (POC) to achieve good physical separation of OC and EC. Another great advantage of using ECT9 to separate OC from EC for isotope analysis (both [13]C & [14]C) is its consistency with the protocol used for OC and EC

concentration measurements. Moreover, the ECT9 method does not require filter samples to be pre-extracted with water before EC analysis (to reduce POC).

**4. Conclusions**

We demonstrate the effectiveness of the ECT9 protocol to physically isolate OC and EC from aerosol samples for [14]C and [13]C analysis by using OC and EC reference materials on their own and as mixtures. It was found that the

ECT9 protocol successfully separates OC and EC fractions with a low (but largely modern) total carbon blank of 1.3±0.6 μg C. The majority (65%) of this extraneous carbon originates from the isolation with the ECT9 protocol, with 35% contributed from graphitization and [14]C measurement of the samples at the KCCAMS facility. After mass balance background corrections, the $FM^{14}C$ results from both bulk pure materials and mixtures (with sample size as small as 5 μgC) can reach the consensus values (Table 2) with an average uncertainty of about 5%.

In addition, we evaluated potential POC formation during ECT9 by investigating thermograms of a variety of reference materials and ambient filter samples. It is demonstrated that ECT9 provides a good alternative for carbonaceous aerosol source apportionment studies, including ultra small sized (5-15 μg C) samples obtained from Arctic regions. To increase the application of isotope data ([14]C or [13]C) in atmospheric research, future efforts should be focused on the comparison on OC/EC separation via different methods/protocols using the same sets of reference





materials. At the same time, the isolation results should be also compared among those methods/protocols widely used in long-term national monitoring network for OC/EC contents, ensuring a consistency in measurements between OC/EC concentrations and their corresponding isotopic compositions.

**Nomenclature**

| | |
|---|---|
| AMS | Accelerator Mass Apectrometry |
| ASTD | Atmospheric Science & Technology Directorate |
| BC | Black carbon |
| CABM | Canadian Aerosol Baseline Measurement |
| CAIR | Carbonaceous Aerosol & Isotope Research |
| CCMR | Climate Chemistry Measurements and Research |
| CC | Carbonate carbon |
| CRD | Climate Research Division |
| EC | Elemental carbon |
| ECCC | Environment and Climate Change Canada |
| ECT9 | EnCan-Total-900 protocol |
| EUSAAR | European Supersites for Atmospheric Aerosol Research |
| FID | Flame ionization detector |
| FM$^{14}$C | Fraction Modern Carbon |
| ICP | Inter-comparison study |
| IRMS | Isotopic Ratio Mass Spectrometer |
| IMPROVE | Interagency Monitoring PROtected Visual Environments |
| KCCAMS | W.M. Keck Carbon Cycle Accelerator Mass Spectrometry Facility |
| MAC | Mass absorption coefficient |
| NIST | National Institute of Standard and Technology |
| OC | Organic carbon |
| PM | Particulate matter |
| POC | Pyrolyzed organic carbon |
| PSAP | Particle Soot Absorption Photometer |
| rBC | Refractory Black Carbon |
| SP2 | Single Particle Soot Photometer |
| SRM | Standard Reference Material |
| TC | Total carbon |
| TEA | Thermal evolution analysis |
| TOA | Thermal optical analysis |
| UCI | University of California, Irvine |



**Data availability**

All data presented in this article are included in the supplement.

**Supplement**

The supplement related to this article is available online at: https://doi.org/10.5194/amt-2020-201-supplement. (to be finalized)

**Author contributions**

Conceptualizing and designing the study: LH, CIC, and GMS

Developing analytical methods and ensuring data quality: LH, GMS, WZ, CIC, BTR

Performing the experiments and data acquisition: WZ, GMS, SRH, VV, BTR

Data organizing /analysis and interpretation: LH, CIC, BTR, GMS, WZ

Writing the paper, including editing and preparing figures and tables: LH, CIC, BTR, GMS, WZ

**Competing interests**

The authors declare that they have no conflicts of interest.

**Acknowledgements**

This research was supported by A-base funding from Environment and Climate Change Canada and the KCCAMS Facility at the University of California, Irvine through G.M.S. We thank D. Enrst (ECCC) and J. Southon (KCCAMS) for supporting $^{13}$C- IRMS and $^{14}$C-AMS analyses, respectively.



**Tables**

**Table 1.** Overview of the bulk reference materials analyzed with the ETC9 method for their total carbon (TC), organic carbon (OC), and elemental carbon (EC) conten

| Reference material | EC | | | | OC | | | | EC + OC mixture | | | |
|---|---|---|---|---|---|---|---|---|---|---|---|---|
| | **Regal black** | | **C1150** | | **Sucrose** | | **Adipic acid** | | **Rice char** | | **SRM-1649a** | |
| | mean | s.d. | mean | s.d. | mean | s.d. | mean | s.d. | mean | s.d. | mean | s.d. |
| TC (%) | 96 | 9 | 98 | 12 | 101[a] | 4 | 43[b] | 5 | 52[c] | 1 | 17.9[d] | 1.1 |
| OC/TC (%) | **3** | 1 | **1** | 2 | **99** | 1 | **100** | 0 | **14** | 1 | **51.5** | 0.8 |
| EC/TC (%) | **97** | 1 | **99** | 2 | **1** | 1 | **0** | 0 | **86** | 1 | **48.5** | 0.8 |
| n | 41 | | 24 | | 117 | | 5 | | 6 | | 6 | |
| Bulk material | fine powder | | | | solution | | fine powder | | | | | |
| Loading method | gravimetric (via a balance with 1 - 0.1 µg accuracy) | | | | volumetric injection | | gravimetric (1 - 0.1 µg accuracy) | | | | | |
| Loading range (µg) | 16 - 134 | | 4 - 104 | | 20 - 80 | | 30 - 250 | | 70 - 210 | | 440 - 1100 | |
| Analysis period | 2015 – 2017 | | 2006, 2013, 2015 | | 2013 - 2018 | | 2015, 2019 | | 2018 | | 2004 - 2005 | |
| Supplier | Aerodyne Research, MA, USA | | McMaster Univ., ON, Canada | | Sigma-Aldrich, MO, USA | | Fisher-Scientific, NH, USA | | Univ. of Zurich, Switzerland | | NIST, MD, USA | |

[a]101% is obtained from the ratio of TC measured to TC calculated from the injected solution of sucrose; [b]49% of TC to bulk material in adipic acid based on its molecul
mass; [c]58.6% of TC to bulk material in Rice char obtained from Hammes et al. (2006); [d]17% of TC to bulk material in SRM 1649a obtained from a critical evaluation o
inter-laboratory data by Currie et al. (2002)




**Table 2.** Overview of the isotopic composition of the reference materials used in this study. Radiocarbon ($^{14}C/^{12}C$, reported as fraction modern (FM$^{14}$C)) was measure at the KCCAMS facility and $\delta^{13}$C at the CAIR lab.

| Reference material | EC | | | | OC | | | | EC + OC mixture | | | |
|---|---|---|---|---|---|---|---|---|---|---|---|---|
| | Regal black | | C1150 | | Sucrose | | Adipic acid | | Rice char | | SRM-1649a | |
| | mean | s.d. | mean | s.d. | mean | s.d. | mean | s.d. | mean | s.d. | mean | s.d. |
| **$^{14}$C analysis** | | | | | | | | | | | | |
| FM$^{14}$C_TC | -0.0001 | 0.0006 | 0.0027 | 0.0008 | 1.0586 | 0.0016 | 0.0000 | 0.0002 | 1.0675 | 0.0007 | 0.5118 | 0.001 |
| n | 2 | | 3 | | 2 | | 5 | | 3 | | 1 | |
| Loading range (µg) | 700 - 750 | | 60 - 560 | | 730 - 770 | | 740 - 1050 | | 900 - 960 | | 760 | |
| CO$_2$ isolation & $^{14}$C/$^{12}$C analysis | Reference material is combusted in 6 mm O.D. quartz tubes with 80 mg CuO for 3 hours at 900°C. Sample-CO$_2$ is purified cryogenically & reduced to graphite (Xu et al., 2007). | | | | | | | | | | | |
| **$\delta^{13}$C analysis** | | | | | | | | | | | | |
| $\delta^{13}$C$_{VPDB}$ (‰) | -27.61 | 0.08 | -23.06 | 0.08 | -12.22 | 0.16 | n/a | | -26.74 | | -25.84 | 0.07 |
| n | 5 | | 5 | | 9 | | n/a | | 1 | | 2 | |
| Loading range (µg or µg C*) | 15 - 70 | | 20 – 50 | | 20 | | n/a | | 160 | | 600 | |
| CO$_2$ isolation | Material is loaded on a quartz filter and combusted in a Sunset OCEC aerosol analyzer (http://www.sunlab.com) using the ECT9 method. Sample-CO$_2$ is collected in a U-shaped flask submerged in liquid N$_2$ at -196°C (Fig. 1b). | | | | | | n/a | | See description for Regal black, C1150, and sucrose. | | | |
| CO$_2$ extraction & $^{13}$C/$^{12}$C analysis | Sample-CO$_2$ is cryogenically purified on a vacuum line and sealed into an ampoule for analysis with a MAT253 Isotopic Ratio Mass Spectrometer (Huang et al., 2013). | | | | | | n/a | | | | | |

*Sucrose was loaded as a solution (µg C), Regal Black, C1150, Adipic acid, Rice char, and SRM-1649a as a fine powder (µg dry mass); n/a = not applicable





**Table 3.** Comparison of the OC and EC ECT9 and Swiss-4S isolation protocols.

| Carrier gas | Carbon fraction | Temperature | Duration | Comments |
|---|---|---|---|---|
| | | °C | s | |
| **ETC9[a]** | | | | |
| He-purge | | 20 – 50 | 90 | Purging of volatile and semi-volatile OC |
| He | OC | 550 | 600 | |
| He | POC + CC | 870 | 600 | Minimizing charred OC contribution to EC |
| $O_2$/He[b] | EC | 900 | 420 | |
| **Swiss-4S[c]** | | | | |
| $O_2$-purge | | 20 – 50 | 90 | Purging of volatile and semi-volatile OC |
| $O_2$ | S1_OC | 375 | 240 | |
| $O_2$ | S2_OC | 475 | 120 | |
| He | S3_OC | 650 | 180 | |
| $O_2$ | S4_EC | 760 | 160 | Water-soluble OC is removed by water extraction prio thermal analysis. |

[a]POC + CC = pyrolysis OC + carbonate carbon; [b]The flow of 10% $O_2$ + 90% He mixing with the flow of 100% He resulting
in 2% $O_2$ + 98%He. in [c]The EC punch is flushed with Milli-Q water prior the analysis to remove the water-soluble OC and
minimize charring (Zhang et al., 2012; Mouteva et al., 2015a).


**Table 4.** Comparison of the procedural contamination with extraneous carbon for aerosol reference materials partitioned into
organic carbon (OC) and elemental carbon (EC) with the ECT9 or Swiss_4S protocols based on their [14]C contents. We assume
a measurement uncertainty of 50% (see Methods).

| Contamination Source | ECT9 | Swiss_4S[a] | |
|---|---|---|---|
| | μg C | 440 | |
| **OC/EC isolation + trapping** | | | |
| Modern | 0.55 | 0.37 | |
| Fossil | 0.30 | 0.13 | |
| Total | 0.85 | 0.50 | |
| **[14]C analysis[b]** | | | |
| Modern | 0.35 | 0.43 | |
| Fossil | 0.10 | 0.53 | 445 |
| Total | 0.45 | 0.97 | |
| **Full set-up** | | | |
| Modern | 0.90 | 0.80 | |
| *Fossil* | *0.40* | *0.67* | |
| *Total* | *1.30* | *1.47* | 450 |

[a]From Mouteva et al. (2015a), [b]Carbon introduced during sample combustion, $CO_2$ purification and graphitization, and
measurement with 14C-AMS.




**Figure captions**

**Figure 1:** Overview of the carbonaceous aerosol analysis system at Environment and Climate Change Canada.
**(a)** Schematic flow chart for $^{13}C$ & $^{14}C$ measurements of OC/EC via ECT9, including 1) OC/EC isolation/$CO_2$
collection via cryo-trapping, 2) $CO_2$ purification, and 3) isotope analysis with IRMS ($^{13}C/^{12}C$ of $CO_2$) or AMS
($^{13}C/^{12}C$ and $^{14}C/^{12}C$ of graphite targets).

**(b)** Thermogram of the ECT9 protocol on a Sunset OC/EC Analyzer. First, organic carbon (OC) is thermally
desorbed at 550ºC for 600 seconds in 100% He, then any pyrolyzed OC (POC), refractory OC, and carbonate carbon
(CC) is released at 870ºC in 100% He for 600 seconds. Finally, elemental carbon (EC) is combusted at 900ºC for 420
seconds by introducing 2% $O_2$ in He. All carbon fractions are oxidized to $CO_2$ followed by reduction to $CH_4$ and
quantification via flame ionization detection (FID) for carbon content or purified and cryo-trapped in Pyrex ampoules
for isotope analysis. Example FID signals are shown for a pure OC reference material (sucrose) mixed with a pure
EC material (regal black) along with the internal standard ($CH_4$).

**Figure 2:** Cross-validation of carbon-mass prepared, isolated by the ECT9 protocol and collected via cryo-trapping at ECCC
and then, retrieved during the purification and graphitization on a KCCAMS vacuum line. Carbon fractions (organic carbon
(OC), elemental carbon (EC), or total carbon (TC)) were isolated from two reference materials for OC (sucrose, adipic acid),
EC (regal black, C1150), and one OC & EC mixture (rice char). Most of the points deviating from the 1:1 line are carbon-rich
reference materials, e.g., Regal black and C1150 (>90% TC), which usually there are greater uncertainties in initial mass
determination via weighing using microbalance, because their sample sizes aimed were very small.

**Figure 3:** Radiocarbon ($^{14}C$) compositions, expressed as Fraction Modern Carbon, of total carbon (TC, circles),
organic carbon (OC, triangles) and elemental carbon (EC, squares) fractions isolated with the ECT9 protocol from
modern or fossil individual reference materials. **a)** Sucrose and **b)** adipic acid are modern and fossil OC, respectively,
**c)** regal black and **d)** C1150 are fossil EC, and **e)** rice char is a mixture of modern OC and EC. Open and solid
symbols represent $^{14}C$ data before and after correction for extraneous carbon introduced during OC/EC isolation and
subsequent $^{14}C$ analysis, respectively. The dashed line indicates the consensus value determined from regular-sized
bulk samples of these materials undergoing off-line combustions (see Table 2).

**Figure 4:** Radiocarbon ($^{14}C$) composition, expressed as Fraction Modern Carbon, of **a)** organic (OC, triangles) or **b)**
elemental (EC, squares) carbon fractions isolated with the ECT9 protocol from mixtures of pure modern OC
(sucrose) with fossil EC (regal black). Open and solid symbols represent $^{14}C$ data before and after correction for
extraneous carbon introduced during OC/EC isolation via ECT9 and subsequent $^{14}C$ analysis via AMS, respectively
(see Table S7). The dashed line indicates the consensus value (see Table 2).

**Figure 5:** Radiocarbon ($^{14}C$) compositions, expressed in fraction modern carbon, of organic (OC, triangles) and
elemental (EC, squares) carbon fractions isolated with the ECT9 protocol from the mixtures of reference materials.
Fraction of modern carbon  **a)** OC and **c)** EC isolated from mixtures of pure fossil OC (adipic acid) with modern bulk
rice char (made of 14% OC and 86 % EC), and of **b)** OC and **d)** EC isolated from mixtures of pure fossil OC (adipic
acid) with modern EC from rice char_EC (rice char _OC has been removed before mixing). Open and solid symbols



represent data before and after correction for extraneous carbon introduced during OC/EC isolation via ECT9 and subsequent [14]C analysis via AMS respectively (Table S7). The dashed line indicates the consensus value (see Table 2).

**Figure 6:** Thermograms of pure or bulk references. **a)** Regal black and **b)** Sucrose and **c)** Rice char. Temperature
(blue solid line) and FID signals (integrated yellow area with green line) on the left axes and laser (red solid line) on the right axis. It is observed that on the three thermograms during the temperature stage of 870°C, the laser transmittance signals decrease first and increases again before the next temperatures stage, minimizing POC fraction, i.e., possible charred OC contribution to EC.

**Figure 7:** Thermograms of the filters directly collected from tailpipe exhaust of a diesel engine vehicle in **a)** and a
gasoline engine passage car in **b)**. The legends are the same as Fig 6. It is noticed that the mass fraction from the temperature stage of 870°C in b) is obvious larger than that in a). The latter is negligible indicating that the amount of POC fraction is sample-matrix dependent. The amount of POC from gasoline vehicle emissions is likely larger than that from diesel vehicle emissions. It was noticed that the laser signal reaches the initial value before the 900°C stage for EC releasing, demonstrating that the charring contribution to EC is minimized.

**Figure 8:** Thermograms of fine particles (PM1.0 μm) from the filter samples collected at an Arctic site, i.e., Alert, NU, Canada in summer **a)** and in winter **b)** of 2015. The legends are the same as Fig 6. It is clearly shown on both thermograms that during 550°C stage, the laser signal starts decreasing (implying charred OC formation) and begins increasing during 870°C and reaches the initial value before the EC stage (indicating the contribution to EC by charred OC is minimized or removed).

**Figure 9:** Thermograms of the SRM 8785 filters (the fine fraction (PM$_{2.5}$) of re-suspended urban dust particles from SRM 1649a and collected on quartz filters) with various amount of materials ranging from 614 mg to1723 mg via two different thermal protocols. **a)** and **b)** were obtained by ECT9. The legends are the same as Fig 6. Both thermograms in a) and b) show the similar patterns as in Fig. 6, 7, 8. that the laser signals reaching the initial value are just before the temperature stage of EC, suggesting that the charred OC contribution to EC is minimized. The
thermogram in **c)** is obtained from the same filter in b) but by Swiss-4 protocol for comparison. The legends are similar except for the integrated area with green line, which stands for $CO_2$ in ppm (by NDIR) instead of FID signals.



**Figures**

Figure 1a . Schematic procedures for $^{13}C$ & $^{14}C$ measurements of OC/EC via ECT9

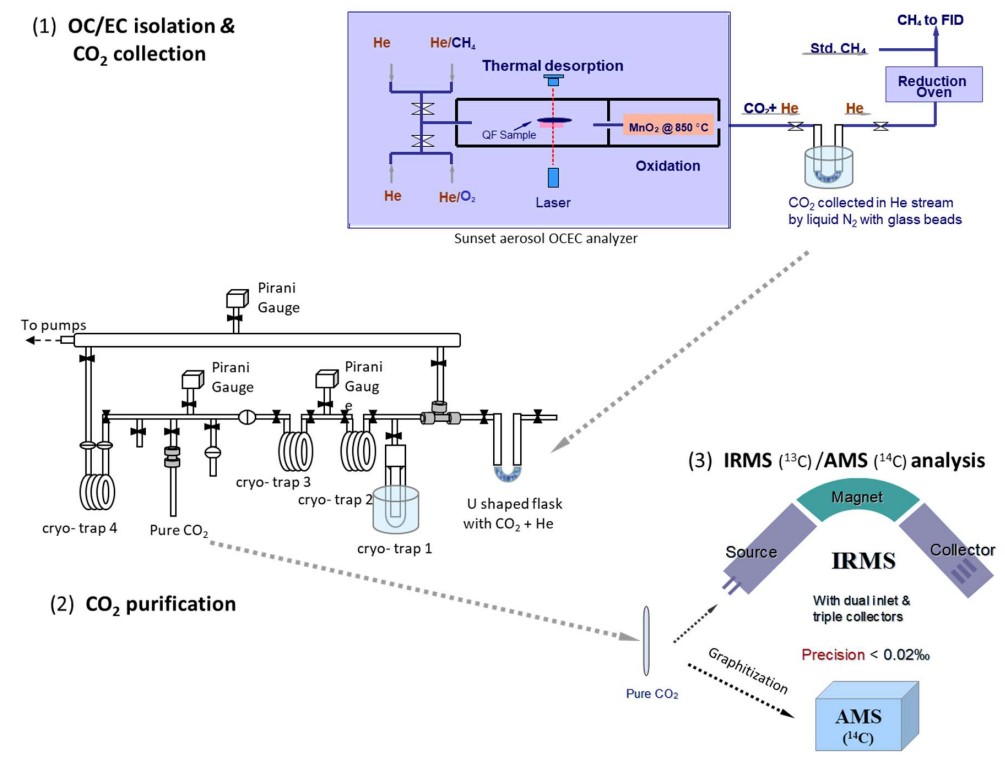




**Figure 1b**

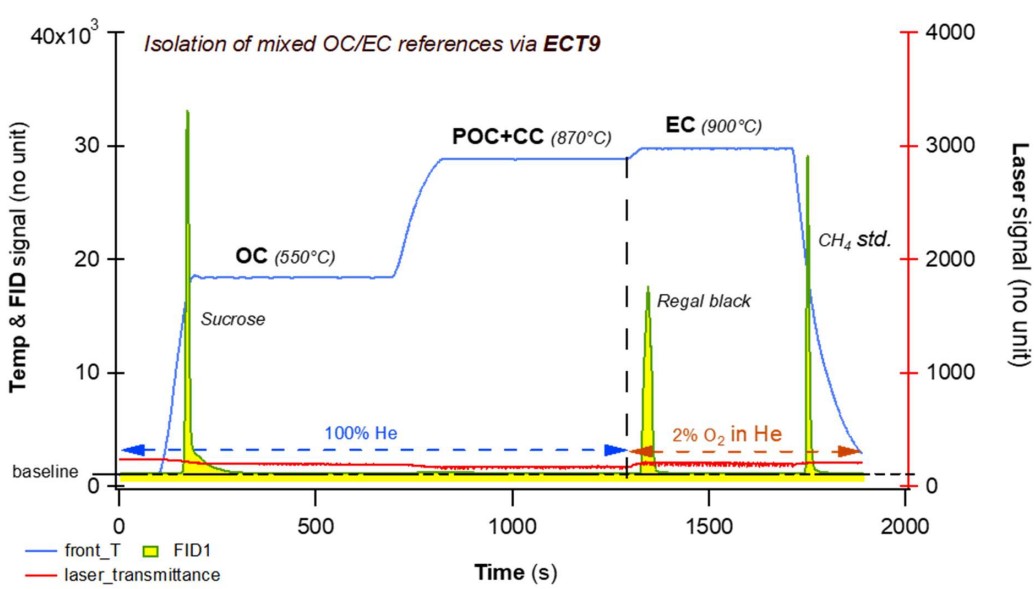






**Figure 2**

a)

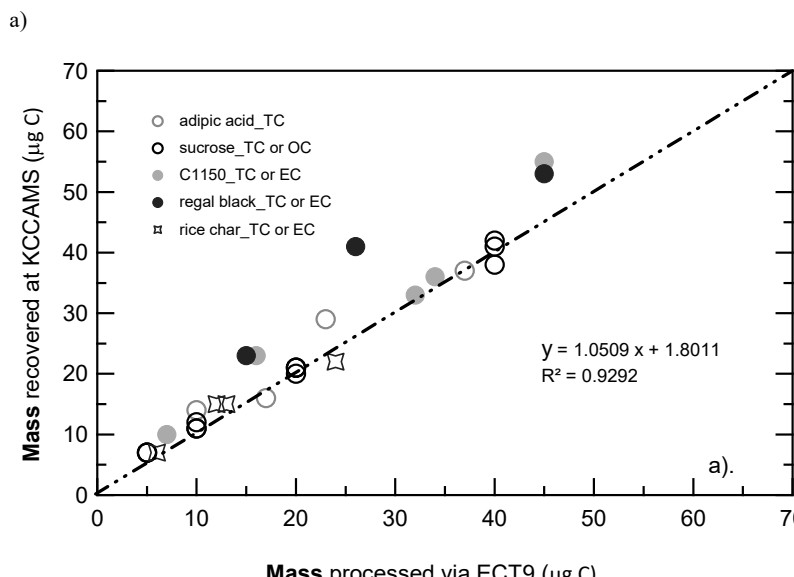


b)

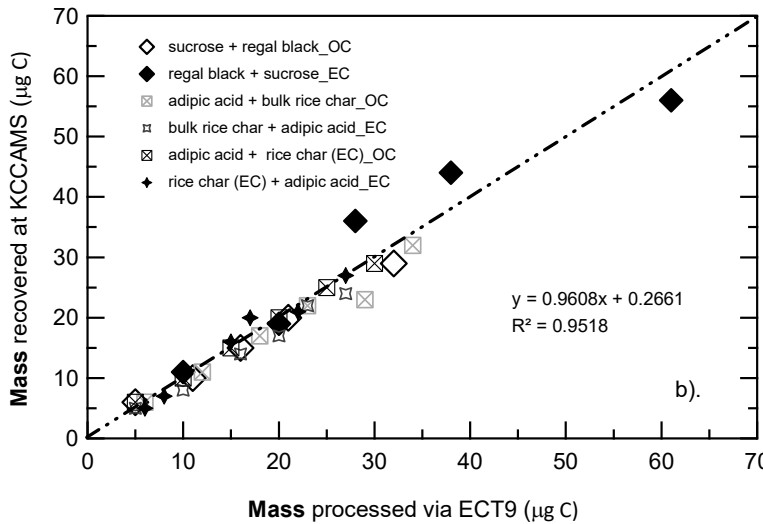






**Figure 3**

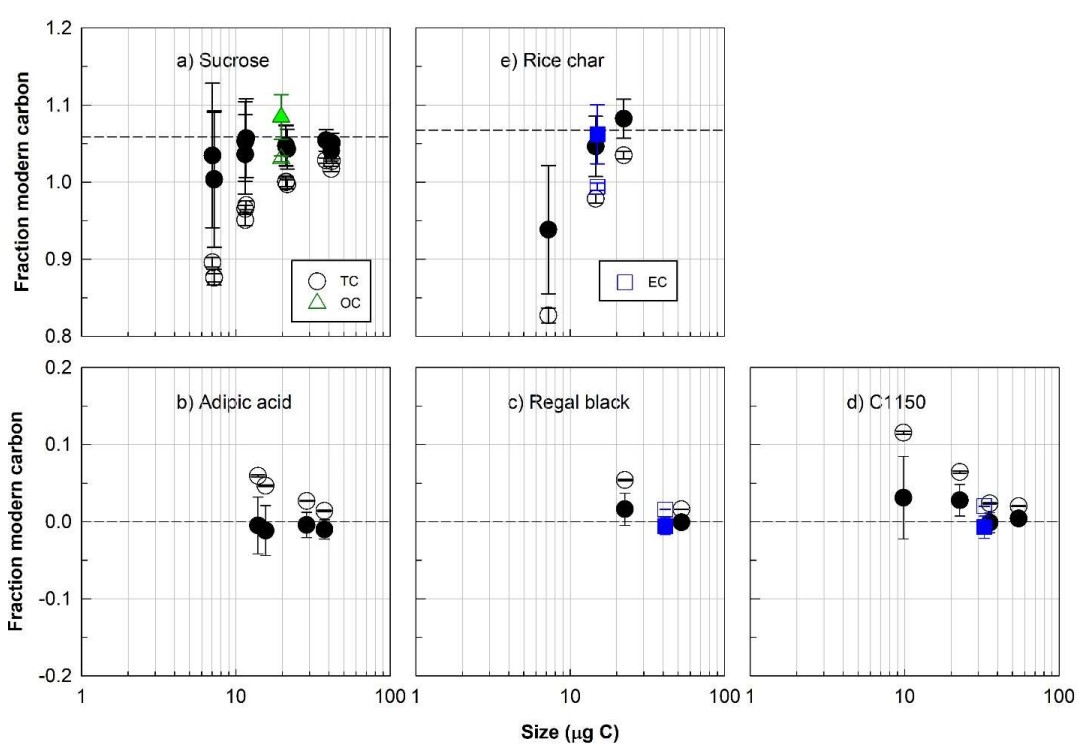








Figure 4

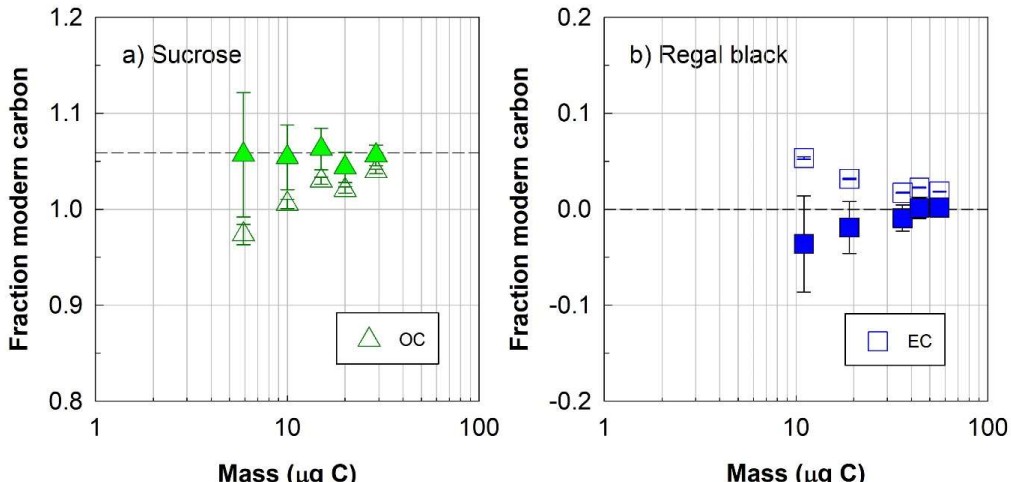




**Figure 5**

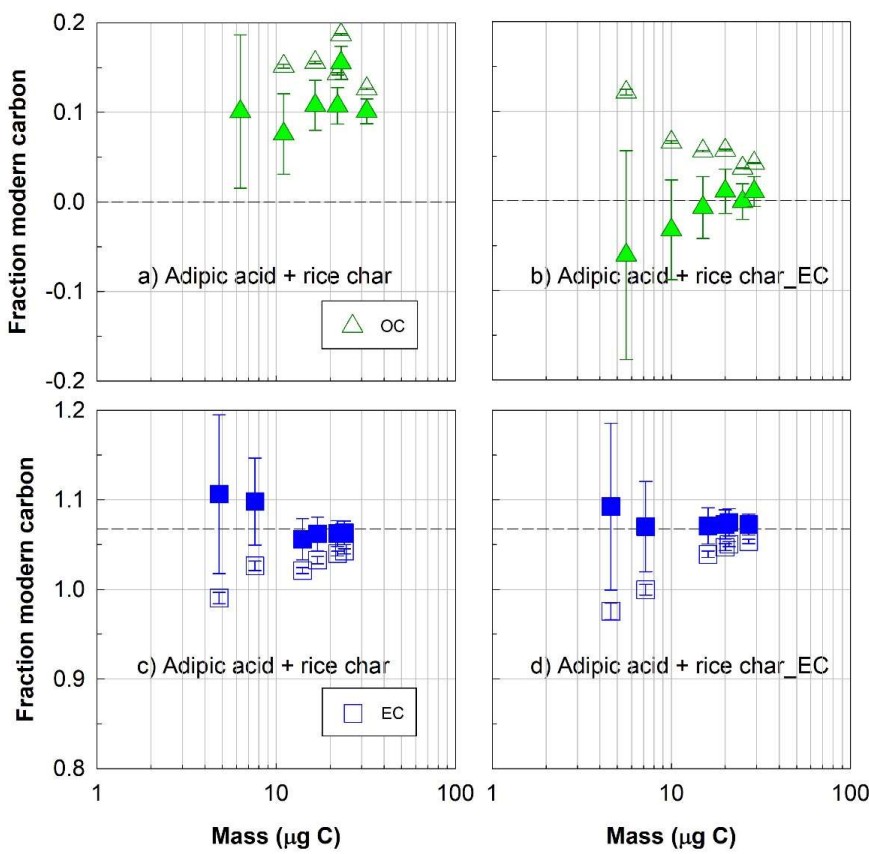






Figure 6

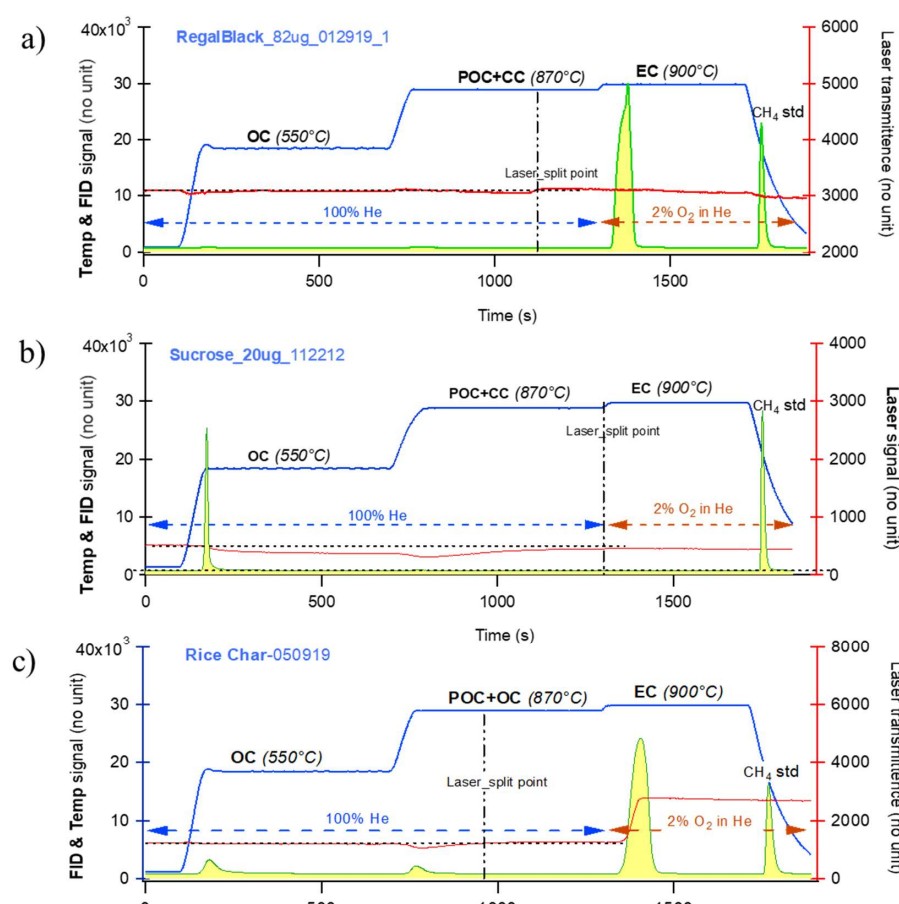









Figure 7

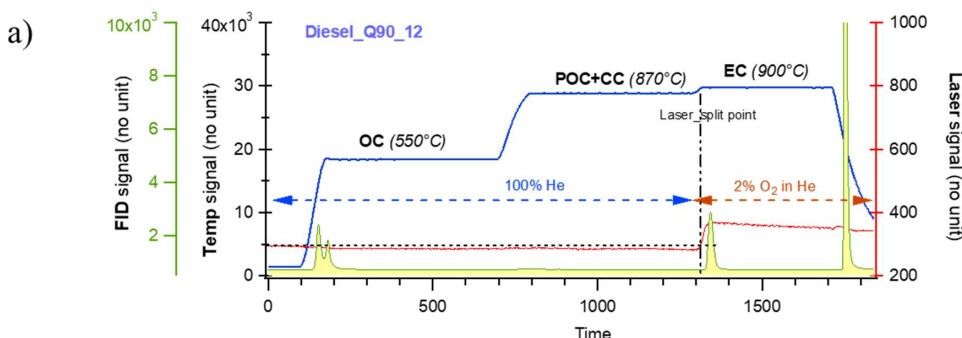

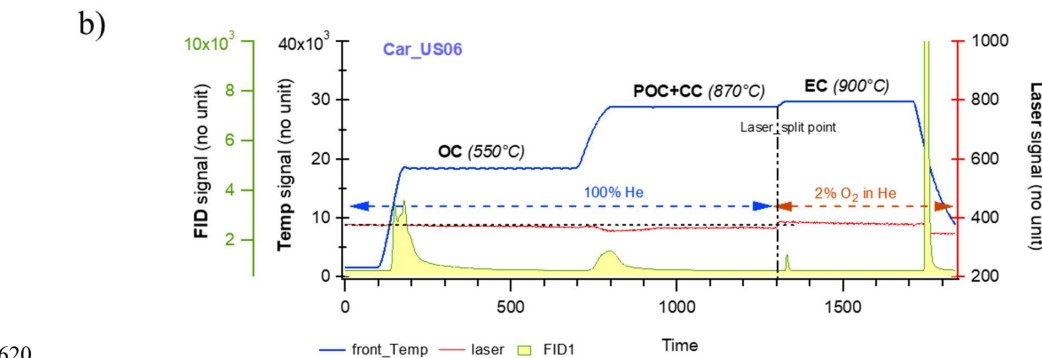









Figure 8

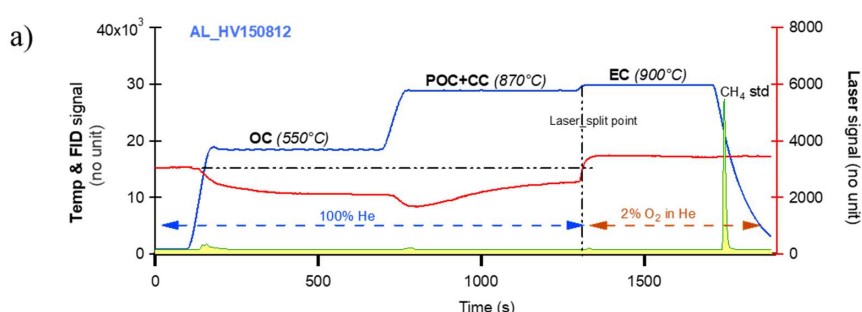

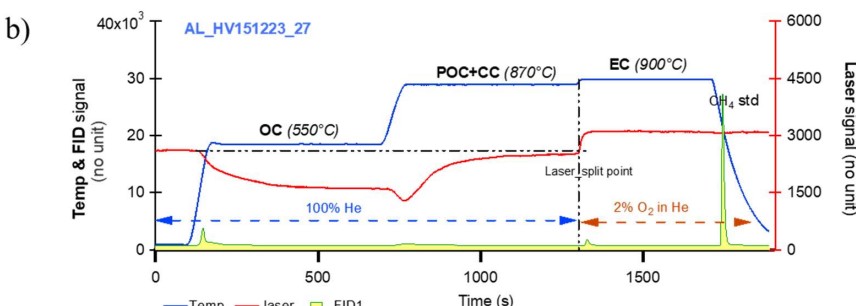








Figure 9

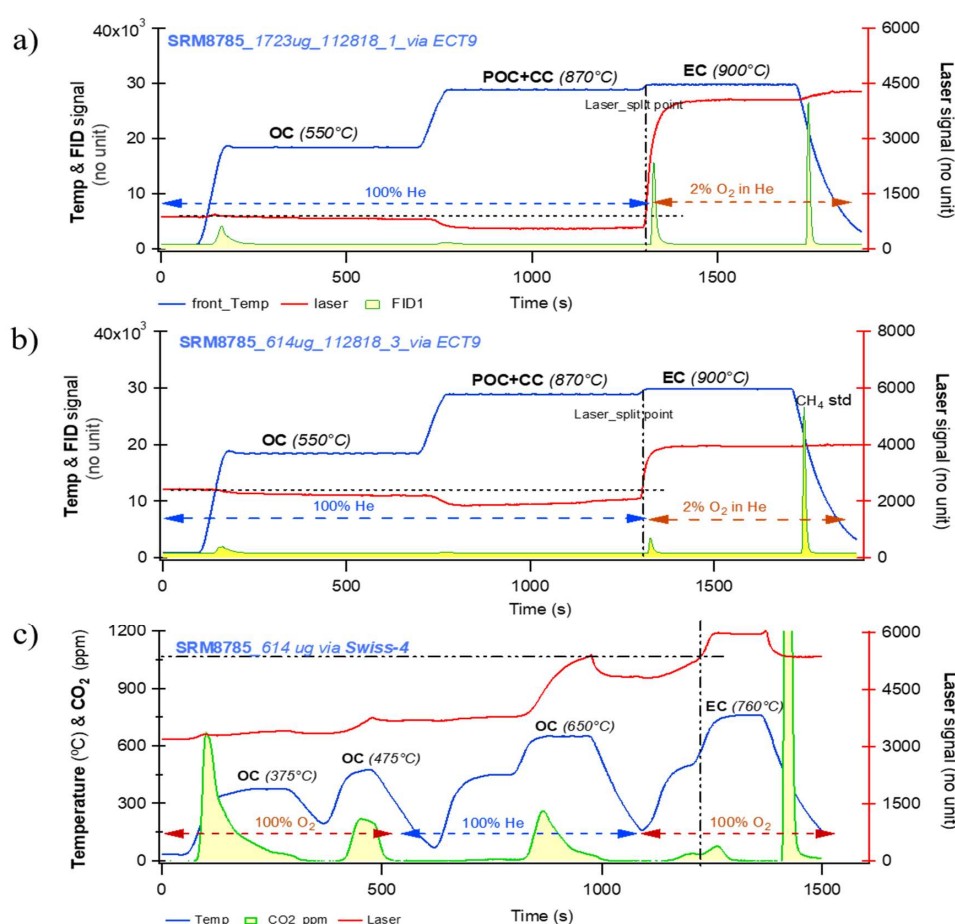



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




**Supplementary Information**

**Table S1.** Individual measurements of OC and EC via ECT9 at ECCC for the references listed in Table 1.

| Lab ID | Date | ᵃLoaded mass on filter | OC | POC+CC | EC | TC | OC$_{total}$/TC | EC/TC | TC/loaded mass |
|---|---|---|---|---|---|---|---|---|---|
| | | μg | | μg/cm² | | | | % | |
| **Regal Black** | (n = 41) | | | | | | | | |
| 16-084-04 | 24-Mar-16 | 24 | 0.02 | 0.69 | 28.34 | 29.05 | 2 | 98 | 121 |
| 16-098-03 | 7-Apr-16 | 22 | -0.05 | 0.48 | 18.63 | 19.06 | 2 | 98 | 87 |
| 16-098-04 | 7-Apr-16 | 23 | 0.43 | 0.91 | 23.89 | 25.23 | 5 | 95 | 110 |
| 16-097-04 | 6-Apr-16 | 19 | 0.44 | 0.48 | 20.02 | 20.94 | 4 | 96 | 110 |
| 16-098-06 | 7-Apr-16 | 18 | 0.15 | 0.49 | 19.05 | 19.69 | 3 | 97 | 109 |
| 17-052-07 | 21-Feb-17 | 21 | 0.17 | 0.50 | 18.42 | 19.09 | 4 | 96 | 91 |
| 17-053-03 | 22-Feb-17 | 16 | 0.14 | 0.76 | 13.24 | 14.14 | 6 | 94 | 88 |
| 17-240-06 | 28-Aug-17 | 18 | 0.27 | 0.59 | 15.12 | 15.98 | 5 | 95 | 90 |
| 17-243-03 | 31-Aug-17 | 20 | 0.00 | 0.42 | 20.22 | 20.64 | 2 | 98 | 104 |
| 17-243-04 | 31-Aug-17 | 24 | 0.14 | 0.20 | 18.79 | 19.13 | 2 | 98 | 79 |
| 15-117-07 | 27-Apr-15 | 30 | 0.22 | 0.95 | 27.46 | 28.63 | 4 | 96 | 95 |
| 16-094-06 | 3-Apr-16 | 32 | 0.80 | 0.76 | 38.23 | 39.79 | 4 | 96 | 124 |
| 16-095-04 | 4-Apr-16 | 27 | 0.39 | 0.57 | 26.11 | 27.07 | 4 | 96 | 100 |
| 16-099-06 | 8-Apr-16 | 27 | 0.03 | 0.87 | 24.68 | 25.58 | 4 | 96 | 95 |
| 16-099-07 | 8-Apr-16 | 26 | 0.14 | 0.95 | 25.37 | 26.46 | 4 | 96 | 102 |
| 17-052-07 | 21-Feb-17 | 25 | 0.12 | 0.92 | 23.47 | 24.51 | 4 | 96 | 98 |
| 15-104-08 | 14-Apr-15 | 52 | 0.00 | 0.85 | 47.21 | 48.06 | 2 | 98 | 92 |
| 16-095-07 | 4-Apr-16 | 47 | 0.30 | 1.18 | 48.19 | 49.67 | 3 | 97 | 106 |
| 16-097-05 | 6-Apr-16 | 43 | 0.32 | 1.03 | 39.78 | 41.13 | 3 | 97 | 96 |
| 16-098-08 | 7-Apr-16 | 50 | 0.12 | 0.67 | 47.38 | 48.17 | 2 | 98 | 96 |
| 17-052-05 | 21-Feb-17 | 53 | 0.90 | 1.74 | 44.31 | 46.95 | 6 | 94 | 89 |
| 17-052-06 | 21-Feb-17 | 42 | 0.22 | 1.37 | 35.51 | 37.10 | 4 | 96 | 88 |
| 17-241-07 | 29-Aug-17 | 44 | 0.52 | 1.51 | 38.78 | 40.81 | 5 | 95 | 93 |





| | | | | | | | | | |
|---|---|---|---|---|---|---|---|---|---|
| 17-241-08 | 29-Aug-17 | 49 | 0.80 | 0.89 | 40.80 | 42.49 | 4 | 96 | 87 |
| 17-243-06 | 31-Aug-17 | 43 | 0.00 | 0.53 | 38.07 | 38.60 | 1 | 99 | 91 |
| 15-117-10 | 27-Apr-15 | 71 | 0.50 | 1.59 | 65.55 | 67.64 | 3 | 97 | 95 |
| 16-098-05 | 7-Apr-16 | 61 | 0.18 | 1.17 | 64.91 | 66.26 | 2 | 98 | 109 |
| 16-099-03 | 8-Apr-16 | 71 | 0.00 | 0.56 | 64.60 | 65.16 | 1 | 99 | 92 |
| 16-099-04 | 8-Apr-16 | 63 | 0.00 | 1.36 | 54.53 | 55.89 | 2 | 98 | 89 |
| 17-052-09 | 21-Feb-17 | 83 | 0.83 | 2.08 | 76.60 | 79.51 | 4 | 96 | 96 |
| 17-243-05 | 21-Feb-17 | 74 | 0.67 | 1.99 | 63.36 | 66.02 | 4 | 96 | 89 |
| 17-243-07 | 31-Aug-17 | 68 | 0.00 | 1.14 | 57.82 | 58.96 | 2 | 98 | 87 |
| 17-243-09 | 31-Aug-17 | 71 | 0.24 | 1.49 | 60.34 | 62.07 | 3 | 97 | 88 |
| 15-117-04 | 27-Apr-15 | 134 | 0.00 | 0.61 | 123.52 | 124.13 | 0 | 100 | 93 |
| 16-098-07 | 7-Apr-16 | 107 | 0.64 | 0.42 | 99.88 | 100.94 | 1 | 99 | 94 |
| 17-240-03 | 28-Aug-17 | 95 | 0.85 | 2.30 | 85.17 | 88.32 | 4 | 96 | 93 |
| 17-241-02 | 29-Aug-17 | 101 | 0.83 | 2.23 | 88.23 | 91.29 | 3 | 97 | 90 |
| 17-241-06 | 29-Aug-17 | 93 | 0.43 | 1.24 | 82.44 | 84.11 | 2 | 98 | 91 |
| 17-240-05 | 28-Aug-17 | 116 | 0.86 | 2.85 | 103.57 | 107.28 | 3 | 97 | 92 |
| 17-243-10 | 31-Aug-17 | 123 | 0.11 | 2.06 | 109.73 | 111.90 | 2 | 98 | 91 |
| 17-244-02 | 1-Sep-17 | 122 | 0.63 | 2.11 | 108.41 | 111.15 | 2 | 98 | 91 |
| | | | | | | mean | 3 | 97 | 96 |
| | | | | | | s.d. | 1 | 1 | 9 |
| **C1150** | (n = 24) | | | | | | | | |
| 06-195-07 | 14-Jul-06 | 4 | 0.05 | 0.05 | 3.17 | 3.26 | 3 | 97 | 81 |
| 06-195-09 | 14-Jul-06 | 7 | 0.23 | 0.00 | 6.35 | 6.57 | 3 | 97 | 94 |
| 06-195-10 | 14-Jul-06 | 10 | 0.48 | 0.18 | 8.91 | 9.57 | 7 | 93 | 96 |
| 06-198-03 | 17-Jul-06 | 18 | 0.12 | 0.09 | 18.18 | 18.39 | 1 | 99 | 102 |
| 06-198-04 | 17-Jul-06 | 25 | 0.32 | 0.42 | 23.22 | 23.96 | 3 | 97 | 96 |
| 06-198-05 | 17-Jul-06 | 42 | 0.02 | 0.32 | 39.75 | 40.08 | 1 | 99 | 95 |
| 06-198-06 | 17-Jul-06 | 34 | 0.48 | 0.42 | 32.94 | 33.84 | 3 | 97 | 100 |
| 06-198-07 | 17-Jul-06 | 15 | 0.26 | 0.41 | 14.45 | 15.11 | 4 | 96 | 101 |
| 13-225-03 | 13-Aug-13 | 25 | 0.00 | 0.00 | 20.29 | 20.29 | 0 | 100 | 80 |
| 13-225-04 | 13-Aug-13 | 89 | 0.28 | 0.00 | 91.34 | 91.62 | 0 | 100 | 102 |
| 13-225-05 | 13-Aug-13 | 30 | 0.00 | 0.00 | 27.50 | 27.50 | 0 | 100 | 93 |
| 13-225-06 | 13-Aug-13 | 46 | 0.00 | 0.00 | 38.35 | 38.35 | 0 | 100 | 84 |
| 13-226-03 | 14-Aug-13 | 10 | 0.05 | 0.01 | 7.33 | 7.39 | 1 | 99 | 78 |



| | | | | | | | | | |
|---|---|---|---|---|---|---|---|---|---|
| 13-226-04 | 14-Aug-13 | 79 | 0.06 | 0.00 | 68.51 | 68.57 | 0 | 100 | 87 |
| 13-226-05 | 14-Aug-13 | 14 | 0.05 | 0.00 | 13.73 | 13.78 | 0 | 100 | 98 |
| 13-226-06 | 14-Aug-13 | 17 | 0.11 | 0.00 | 20.16 | 20.27 | 1 | 99 | 116 |
| 13-226-07 | 14-Aug-13 | 49 | 0.16 | 0.03 | 49.37 | 49.56 | 0 | 100 | 101 |
| 15-122-09 | 2-May-15 | 72 | 0.05 | 0.00 | 69.62 | 69.67 | 0 | 100 | 97 |
| 15-122-07 | 2-May-15 | 71 | 0.19 | 0.00 | 70.79 | 70.98 | 0 | 100 | 100 |
| 15-122-08 | 2-May-15 | 104 | 0.00 | 0.00 | 97.78 | 97.78 | 0 | 100 | 94 |
| 15-123-03 | 3-May-15 | 22 | 0.13 | 0.00 | 29.38 | 29.51 | 0 | 100 | 134 |
| 15-123-04 | 3-May-15 | 71 | 0.00 | 0.00 | 76.06 | 76.06 | 0 | 100 | 107 |
| 15-123-05 | 3-May-15 | 27 | 0.32 | 0.00 | 25.96 | 26.28 | 1 | 99 | 97 |
| 15-123-06 | 3-May-15 | 59 | 0.11 | 0.15 | 66.40 | 66.66 | 0 | 99 | 113 |
| | | | | | | **mean** | **1** | **99** | **98** |
| | | | | | | **s.d.** | **2** | **2** | **12** |
| **Sucrose** | (n = 117) | | | | | | | | |
| 13-332-02 | 28-Nov-13 | 20 | 19.76 | 0.35 | 0.00 | 20.11 | 100 | 0 | 101 |
| 13-332-03 | 28-Nov-13 | 20 | 19.77 | 0.48 | 0.02 | 20.27 | 100 | 0 | 101 |
| 13-333-02 | 28-Nov-13 | 20 | 19.46 | 0.44 | 0.00 | 19.90 | 100 | 0 | 100 |
| 13-332-08 | 28-Nov-13 | 40 | 37.50 | 1.00 | 0.00 | 38.50 | 100 | 0 | 96 |
| 13-332-10 | 28-Nov-13 | 40 | 38.77 | 0.98 | 0.00 | 39.75 | 100 | 0 | 99 |
| 13-333-03 | 29-Nov-13 | 40 | 39.51 | 1.11 | 0.01 | 40.63 | 100 | 0 | 102 |
| 13-333-05 | 29-Nov-13 | 80 | 75.63 | 1.73 | 0.22 | 77.58 | 100 | 0 | 97 |
| 13-333-08 | 29-Nov-13 | 80 | 74.25 | 2.14 | 0.07 | 76.46 | 100 | 0 | 96 |
| 13-333-07 | 29-Nov-13 | 80 | 76.43 | 2.05 | 0.07 | 78.55 | 100 | 0 | 98 |
| 14-129-02 | 9-May-14 | 20 | 19.39 | 0.29 | 0.06 | 19.74 | 100 | 0 | 99 |
| 14-129-03 | 9-May-14 | 20 | 19.33 | 0.16 | 0.05 | 19.54 | 100 | 0 | 98 |
| 14-132-02 | 12-May-14 | 20 | 19.71 | 0.00 | 0.00 | 19.71 | 100 | 0 | 99 |
| 14-133-03 | 13-May-14 | 40 | 39.16 | 0.66 | 0.60 | 40.42 | 99 | 1 | 101 |
| 14-133-04 | 13-May-14 | 40 | 39.67 | 0.53 | 0.10 | 40.30 | 100 | 0 | 101 |
| 14-134-02 | 14-May-14 | 40 | 39.44 | 0.31 | 0.11 | 39.86 | 100 | 0 | 100 |
| 14-134-03 | 14-May-14 | 80 | 80.11 | 0.80 | 0.10 | 81.01 | 100 | 0 | 101 |
| 14-134-04 | 14-May-14 | 80 | 79.39 | 1.01 | 0.36 | 80.76 | 100 | 0 | 101 |
| 14-134-05 | 14-May-14 | 80 | 78.49 | 1.86 | 1.46 | 81.81 | 98 | 2 | 102 |
| 14-231-02 | 19-Aug-14 | 20 | 19.03 | 0.28 | 0.12 | 19.43 | 99 | 1 | 97 |
| 14-234-02 | 22-Aug-14 | 20 | 19.20 | 0.50 | 0.13 | 19.83 | 99 | 1 | 99 |





| 14-235-02 | 23-Aug-14 | 20 | 19.06 | 0.55 | 0.00 | 19.61 | 100 | 0 | 98 |
| 14-233-05 | 21-Aug-14 | 40 | 38.76 | 0.99 | 0.20 | 39.95 | 99 | 1 | 100 |
| 14-233-06 | 21-Aug-14 | 40 | 38.22 | 0.00 | 0.00 | 38.22 | 100 | 0 | 96 |
| 14-233-07 | 21-Aug-14 | 40 | 38.32 | 0.04 | 0.00 | 38.36 | 100 | 0 | 96 |
| 14-235-08 | 23-Aug-14 | 80 | 78.25 | 1.44 | 0.18 | 79.87 | 100 | 0 | 100 |
| 14-235-09 | 23-Aug-14 | 80 | 79.46 | 0.27 | 0.00 | 79.73 | 100 | 0 | 100 |
| 14-238-04 | 26-Aug-14 | 80 | 76.15 | 1.47 | 0.38 | 78.00 | 100 | 0 | 98 |
| 15-015-03 | 15-Jan-15 | 20 | 18.67 | 1.22 | 0.10 | 19.99 | 99 | 1 | 100 |
| 15-015-04 | 15-Jan-15 | 20 | 18.65 | 1.51 | 0.18 | 20.34 | 99 | 1 | 102 |
| 15-019-02 | 19-Jan-15 | 20 | 18.95 | 1.01 | 0.01 | 19.97 | 100 | 0 | 100 |
| 15-019-03 | 19-Jan-15 | 40 | 35.12 | 2.62 | 1.07 | 38.81 | 97 | 3 | 97 |
| 15-020-02 | 20-Jan-15 | 40 | 36.63 | 1.84 | 0.17 | 38.64 | 100 | 0 | 97 |
| 15-020-05 | 20-Jan-15 | 40 | 37.43 | 2.43 | 0.29 | 40.15 | 99 | 1 | 100 |
| 15-020-06 | 20-Jan-15 | 80 | 75.34 | 3.27 | 0.87 | 79.48 | 99 | 1 | 99 |
| 15-020-07 | 20-Jan-15 | 80 | 76.30 | 3.42 | 0.92 | 80.64 | 99 | 1 | 101 |
| 15-020-08 | 20-Jan-15 | 80 | 76.65 | 2.85 | 0.72 | 80.22 | 99 | 1 | 100 |
| 15-097-03 | 10-Apr-15 | 20 | 19.79 | 0.41 | 0.00 | 20.20 | 100 | 0 | 101 |
| 15-114-02 | 27-Apr-15 | 20 | 17.15 | 2.41 | 0.12 | 19.68 | 99 | 1 | 98 |
| 15-108-02 | 21-Apr-15 | 20 | 18.62 | 1.28 | 0.00 | 19.90 | 100 | 0 | 100 |
| 15-097-04 | 10-Apr-15 | 40 | 39.35 | 0.85 | 0.02 | 40.22 | 100 | 0 | 101 |
| 15-097-05 | 10-Apr-15 | 40 | 38.90 | 1.80 | 1.02 | 41.72 | 98 | 2 | 104 |
| 15-097-06 | 10-Apr-15 | 40 | 38.59 | 1.75 | 0.88 | 41.22 | 98 | 2 | 103 |
| 15-108-04 | 21-Apr-15 | 80 | 76.10 | 4.20 | 0.23 | 80.53 | 100 | 0 | 101 |
| 15-108-03 | 21-Apr-15 | 80 | 76.47 | 4.13 | 0.31 | 80.91 | 100 | 0 | 101 |
| 15-108-06 | 21-Apr-15 | 80 | 74.94 | 4.89 | 0.70 | 80.53 | 99 | 1 | 101 |
| 15-280-03 | 8-Oct-15 | 20 | 17.56 | 2.64 | 0.04 | 20.24 | 100 | 0 | 101 |
| 15-280-04 | 8-Oct-15 | 20 | 17.34 | 2.95 | 0.05 | 20.34 | 100 | 0 | 102 |
| 15-280-05 | 8-Oct-15 | 20 | 16.99 | 3.00 | 0.00 | 19.99 | 100 | 0 | 100 |
| 15-287-02 | 14-Oct-15 | 40 | 34.13 | 4.64 | 0.13 | 38.90 | 100 | 0 | 97 |
| 15-287-04 | 14-Oct-15 | 40 | 34.72 | 4.81 | 0.15 | 39.68 | 100 | 0 | 99 |
| 15-288-03 | 15-Oct-15 | 40 | 33.67 | 4.98 | 0.17 | 38.82 | 100 | 0 | 97 |
| 15-292-03 | 19-Oct-15 | 80 | 70.58 | 6.94 | 1.31 | 78.83 | 98 | 2 | 99 |
| 15-292-04 | 19-Oct-15 | 80 | 69.29 | 7.36 | 1.53 | 78.18 | 98 | 2 | 98 |
| 15-292-05 | 19-Oct-15 | 80 | 69.29 | 7.23 | 1.47 | 77.99 | 98 | 2 | 97 |


| 16-026-03 | 26-Jan-16 | 20 | 17.74 | 2.70 | 0.02 | 20.46 | 100 | 0 | 102 |
| 16-026-05 | 26-Jan-16 | 20 | 16.85 | 3.37 | 0.12 | 20.34 | 99 | 1 | 102 |
| 16-027-05 | 27-Jan-16 | 20 | 16.68 | 3.24 | 0.10 | 20.02 | 100 | 0 | 100 |
| 16-026-06 | 26-Jan-16 | 40 | 34.15 | 4.79 | 0.18 | 39.12 | 100 | 0 | 98 |
| 16-027-04 | 27-Jan-16 | 40 | 33.69 | 4.98 | 0.51 | 39.18 | 99 | 1 | 98 |
| 16-027-06 | 27-Jan-16 | 40 | 33.14 | 5.39 | 0.75 | 39.28 | 98 | 2 | 98 |
| 16-027-07 | 27-Jan-16 | 80 | 69.99 | 7.15 | 2.28 | 79.42 | 97 | 3 | 99 |
| 16-028-03 | 28-Jan-16 | 80 | 71.40 | 7.34 | 1.98 | 80.72 | 98 | 2 | 101 |
| 16-028-04 | 28-Jan-16 | 80 | 71.87 | 7.06 | 1.91 | 80.84 | 98 | 2 | 101 |
| 16-243-03 | 30-Aug-16 | 20 | 16.69 | 3.24 | 0.65 | 20.58 | 97 | 3 | 103 |
| 16-243-04 | 30-Aug-16 | 20 | 17.35 | 3.35 | 0.07 | 20.77 | 100 | 0 | 104 |
| 16-244-02 | 31-Aug-16 | 20 | 16.80 | 2.92 | 0.85 | 20.57 | 96 | 4 | 103 |
| 16-244-05 | 31-Aug-16 | 40 | 35.61 | 3.87 | 1.26 | 40.74 | 97 | 3 | 102 |
| 16-244-06 | 31-Aug-16 | 40 | 35.76 | 3.87 | 1.29 | 40.92 | 97 | 3 | 102 |
| 16-244-07 | 31-Aug-16 | 40 | 35.81 | 4.20 | 1.85 | 41.86 | 96 | 4 | 105 |
| 16-250-02 | 6-Sep-16 | 80 | 77.54 | 3.94 | 1.34 | 82.82 | 98 | 2 | 104 |
| 16-250-03 | 6-Sep-16 | 80 | 77.77 | 3.81 | 1.26 | 82.84 | 98 | 2 | 104 |
| 16-250-04 | 6-Sep-16 | 80 | 77.95 | 3.81 | 1.25 | 83.01 | 98 | 2 | 104 |
| 17-038-04 | 7-Feb-17 | 20 | 14.57 | 4.14 | 0.93 | 19.64 | 95 | 5 | 98 |
| 17-039-02 | 8-Feb-17 | 20 | 14.99 | 3.88 | 0.84 | 19.71 | 96 | 4 | 99 |
| 17-039-03 | 8-Feb-17 | 20 | 14.74 | 4.31 | 0.79 | 19.84 | 96 | 4 | 99 |
| 17-039-04 | 8-Feb-17 | 40 | 32.68 | 5.47 | 1.20 | 39.35 | 97 | 3 | 98 |
| 17-039-05 | 8-Feb-17 | 40 | 34.09 | 5.70 | 1.00 | 40.79 | 98 | 2 | 102 |
| 17-039-06 | 8-Feb-17 | 40 | 33.22 | 5.89 | 2.47 | 41.58 | 94 | 6 | 104 |
| 17-041-02 | 10-Feb-17 | 80 | 74.47 | 7.17 | 1.59 | 83.23 | 98 | 2 | 104 |
| 17-041-03 | 10-Feb-17 | 80 | 73.71 | 5.02 | 1.61 | 80.34 | 98 | 2 | 100 |
| 17-041-05 | 10-Feb-17 | 80 | 70.96 | 8.04 | 2.31 | 81.31 | 97 | 3 | 102 |
| 18-037-03 | 6-Feb-18 | 20 | 20.31 | 0.00 | 0.21 | 20.52 | 99 | 1 | 103 |
| 18-032-04 | 1-Feb-18 | 20 | 20.06 | 0.00 | 0.11 | 20.17 | 99 | 1 | 101 |
| 18-036-03 | 5-Feb-18 | 20 | 20.01 | 0.00 | 0.16 | 20.17 | 99 | 1 | 101 |
| 18-033-06 | 2-Feb-18 | 40 | 37.87 | 1.66 | 1.71 | 41.24 | 96 | 4 | 103 |
| 18-037-04 | 6-Feb-18 | 40 | 39.36 | 1.17 | 1.39 | 41.92 | 97 | 3 | 105 |
| 18-037-08 | 6-Feb-18 | 40 | 39.02 | 1.30 | 1.58 | 41.90 | 96 | 4 | 105 |
| 18-037-09 | 6-Feb-18 | 80 | 73.37 | 2.92 | 2.15 | 78.44 | 97 | 3 | 98 |



| | | | | | | | | |
|---|---|---|---|---|---|---|---|---|
| 18-037-10 | 6-Feb-18 | 80 | 74.15 | 3.26 | 2.33 | 79.74 | 97 | 3 | 100 |
| 18-037-11 | 6-Feb-18 | 80 | 73.89 | 2.90 | 2.07 | 78.86 | 97 | 3 | 99 |
| 18-129-02 | 9-May-18 | 20 | 19.45 | 0.37 | 0.03 | 19.85 | 100 | 0 | 99 |
| 18-129-03 | 9-May-18 | 20 | 19.81 | 0.35 | 0.07 | 20.23 | 100 | 0 | 101 |
| 18-129-04 | 9-May-18 | 20 | 20.06 | 0.59 | 0.14 | 20.79 | 99 | 1 | 104 |
| 18-129-05 | 9-May-18 | 40 | 38.26 | 1.72 | 1.19 | 41.17 | 97 | 3 | 103 |
| 18-129-06 | 9-May-18 | 40 | 40.03 | 1.37 | 0.79 | 42.19 | 98 | 2 | 105 |
| 18-129-07 | 9-May-18 | 40 | 38.42 | 1.79 | 1.21 | 41.42 | 97 | 3 | 104 |
| 18-130-02 | 10-May-18 | 80 | 80.93 | 0.95 | 0.58 | 82.46 | 99 | 1 | 103 |
| 18-130-03 | 10-May-18 | 80 | 81.34 | 1.02 | 0.28 | 82.64 | 100 | 0 | 103 |
| 18-131-08 | 11-May-18 | 80 | 81.52 | 1.86 | 0.72 | 84.10 | 99 | 1 | 105 |
| 18-297-02 | 24-Oct-18 | 20 | 19.44 | 0.45 | 0.03 | 19.92 | 100 | 0 | 100 |
| 18-302-03 | 29-Oct-18 | 20 | 19.09 | 0.84 | 0.36 | 20.29 | 98 | 2 | 101 |
| 18-298-06 | 25-Oct-18 | 20 | 19.17 | 0.71 | 0.13 | 20.01 | 99 | 1 | 100 |
| 18-302-06 | 29-Oct-18 | 40 | 39.37 | 0.89 | 0.17 | 40.43 | 100 | 0 | 101 |
| 18-298-07 | 25-Oct-18 | 40 | 39.68 | 0.42 | 0.29 | 40.39 | 99 | 1 | 101 |
| 18-309-08 | 5-Nov-18 | 40 | 41.22 | 0.25 | 0.00 | 41.47 | 100 | 0 | 104 |
| 18-309-11 | 5-Nov-18 | 80 | 78.46 | 0.81 | 0.20 | 79.47 | 100 | 0 | 99 |
| 18-309-14 | 5-Nov-18 | 80 | 78.26 | 1.56 | 0.15 | 79.97 | 100 | 0 | 100 |
| 18-310-03 | 6-Nov-18 | 80 | 82.01 | 1.75 | 0.50 | 84.26 | 99 | 1 | 105 |
| 18-355-02 | 21-Dec-18 | 20 | 22.04 | 0.99 | 0.24 | 23.27 | 99 | 1 | 116 |
| 18-355-03 | 21-Dec-18 | 20 | 21.48 | 1.11 | 0.15 | 22.74 | 99 | 1 | 114 |
| 18-355-04 | 21-Dec-18 | 20 | 21.17 | 1.31 | 0.26 | 22.74 | 99 | 1 | 114 |
| 18-361-04 | 27-Dec-18 | 40 | 42.02 | 1.29 | 0.13 | 43.44 | 100 | 0 | 109 |
| 18-361-05 | 27-Dec-18 | 40 | 41.56 | 0.93 | 0.00 | 42.49 | 100 | 0 | 106 |
| 18-361-06 | 27-Dec-18 | 40 | 41.06 | 1.85 | 0.23 | 43.14 | 99 | 1 | 108 |
| 18-361-07 | 27-Dec-18 | 80 | 85.76 | 2.27 | 0.75 | 88.78 | 99 | 1 | 111 |
| 18-361-08 | 27-Dec-18 | 80 | 86.49 | 2.48 | 0.78 | 89.75 | 99 | 1 | 112 |
| 18-361-09 | 27-Dec-18 | 80 | 85.98 | 2.63 | 0.61 | 89.22 | 99 | 1 | 112 |
| | | | | | | **mean** | **99** | **1** | **101** |
| | | | | | | **s.d.** | **1** | **1** | **4** |
| **Adipic Acid** | (n = 5) | | | | | | | | |
| 15-062-06 | 3-Mar-15 | 34 | 13.67 | 0.09 | 0.00 | 13.76 | 100 | 0 | 40 |
| 15-062-05 | 3-Mar-15 | 102 | 47.47 | 0.00 | 0.00 | 47.47 | 100 | 0 | 47 |





| 15-100-02 | 13-Apr-15 | n/a | 5.25 | 0.00 | 0.05 | 5.30 | 99 | 1 | n/a |
|---|---|---|---|---|---|---|---|---|---|
| 19-137-05 | 17-May-19 | 253 | 120.68 | 1.05 | 0.07 | 121.80 | 100 | 0 | 48 |
| 19-137-06 | 17-May-19 | 28 | 10.62 | 0.00 | 0.00 | 10.62 | 100 | 0 | 38 |
| | | | | | | **mean** | **100** | **0** | **43** |
| | | | | | | **s.d.** | **0** | **0** | **5** |
| **Rice Char** | (n = 6) | | | | | | | | |
| 18-158-05 | 7-Jun-18 | 112 | 4.27 | 3.25 | 49.54 | 57.06 | 13 | 87 | 51 |
| 18-164-05 | 13-Jun-18 | 212 | 8.73 | 6.22 | 96.87 | 111.82 | 13 | 87 | 53 |
| 18-165-06 | 14-Jun-18 | 79 | 2.96 | 2.86 | 35.46 | 41.28 | 14 | 86 | 52 |
| 18-169-04 | 18-Jun-18 | 71 | 2.76 | 2.70 | 30.18 | 35.64 | 15 | 85 | 51 |
| 18-172-05 | 21-Jun-18 | 150 | 5.74 | 4.28 | 70.40 | 80.42 | 12 | 88 | 54 |
| 18-176-06 | 25-Jun-18 | 121 | 4.83 | 4.67 | 56.43 | 65.93 | 14 | 86 | 54 |
| | | | | | | **mean** | **14** | **86** | **52** |
| | | | | | | **s.d.** | **1** | **1** | **1** |
| **SRM-1649a** | (n = 6) | | | | | | | | |
| 04-271-04 | 27-Sep-04 | 690 | 29.94 | 9.65 | 36.46 | 76.05 | 52.1 | 47.9 | 16.5 |
| 04-322-10 | 17-Nov-04 | 490 | 25.82 | 7.18 | 30.41 | 63.41 | 52.0 | 48.0 | 19.4 |
| 04-322-12 | 17-Nov-04 | 880 | 40.28 | 11.25 | 47.71 | 99.24 | 51.9 | 48.1 | 16.9 |
| 05-046-02 | 15-Feb-05 | 1101 | 51.66 | 16.59 | 67.16 | 135.41 | 50.4 | 49.6 | 18.5 |
| 05-046-03 | 15-Feb-05 | 441 | 21.06 | 6.41 | 25.35 | 52.82 | 52.0 | 48.0 | 18.0 |
| 05-046-04 | 15-Feb-05 | 855 | 40.33 | 12.37 | 51.22 | 103.92 | 50.7 | 49.3 | 18.2 |
| | | | | | | **mean** | **51.5** | **48.5** | **17.9** |
| | | | | | | **s.d.** | **0.8** | **0.8** | **1.1** |

[a]Loaded mass are the weighed mass (for Regal black, C1150, Adipic acid, Rice char and SRM-1649a) or injected mass (sucrose) on the filter.





**Table S2.** Radiocarbon content of bulk reference materials, expressed as fraction modern carbon (FM) with and without background correction. $CO_2$ isolation and $^{14}C/^{12}C$ analysis were carried out at KCCAMS, UCI (the method is described in Table 2).

| UCI AMS # | Size | Corrected FM | | Uncorrected FM | |
|---|---|---|---|---|---|
| | µg C | | ± | | ± |
| **Sucrose** | | | | | |
| 150230 | 735 | 1.0597 | 0.0021 | 1.0597 | 0.0021 |
| 150231 | 769 | 1.0575 | 0.0017 | 1.0574 | 0.0017 |
| **AdipicAcid** | | | | | |
| 123428 | 876 | 0.0002 | 0.0005 | 0.0020 | 0.0001 |
| 123430 | 851 | 0.0001 | 0.0005 | 0.0019 | 0.0001 |
| 123431 | 934 | -0.0001 | 0.0005 | 0.0016 | 0.0001 |
| 123432 | 1053 | -0.0003 | 0.0005 | 0.0015 | 0.0001 |
| 123433 | 740 | -0.0001 | 0.0005 | 0.0016 | 0.0001 |
| **Regal Black** | | | | | |
| 150228 | 717 | 0.0004 | 0.0005 | 0.0019 | 0.0001 |
| 150229 | 752 | -0.0005 | 0.0005 | 0.0011 | 0.0000 |
| **C1150** | | | | | |
| 150232 | 88 | 0.0026 | 0.0005 | 0.0042 | 0.0001 |
| 150233 | 64 | 0.0035 | 0.0005 | 0.0050 | 0.0002 |
| 150234 | 560 | 0.0019 | 0.0005 | 0.0035 | 0.0001 |
| **RiceChar** | | | | | |
| 123434 | 924 | 1.0683 | 0.0023 | 1.0683 | 0.0023 |
| 123435 | 913 | 1.0670 | 0.0018 | 1.0670 | 0.0018 |
| 123436 | 961 | 1.0673 | 0.0019 | 1.0672 | 0.0019 |



**Table S3.** Stable isotopic composition ($^{13}C/^{12}C$) of OC and EC fractions or bulk materials. $CO_2$ isolation and $^{13}C/^{12}C$ analysis were carried out at the CA
925  lab, CRD, ASTD/ECCC (the method is described in Table 2).

| Reference m | Lab ID | Date | Fraction | Loaded mass on filter | $\delta^{13}C_{VPDB}$ |
|---|---|---|---|---|---|
| | | | | µg or µg C[a] | ‰ |
| **Regal Black** | 16-036-04 | 5-Feb-16 | EC | 16 | -27.67 |
| (n = 5) | 16-036-05 | 5-Feb-16 | EC | 27 | -27.49 |
| | 16-036-06 | 5-Feb-16 | EC | 22 | -27.67 |
| | 16-036-08 | 5-Feb-16 | EC | 59 | -27.62 |
| | 16-036-09 | 5-Feb-16 | EC | 68 | -27.57 |
| | | | | **mean** | **-27.61** |
| | | | | **s.d.** | **0.08** |
| **C1150** | 13-013-05 | 13-Jan-13 | EC | 50 | -23.01 |
| (n = 5) | 13-013-07 | 13-Jan-13 | EC | 22 | -23.16 |
| | 13-013-08 | 13-Jan-13 | EC | 48 | -22.96 |
| | 16-036-06 | 5-Feb-16 | EC | 30 | -23.14 |
| | 16-036-07 | 5-Feb-16 | EC | 46 | -23.05 |
| | | | | **mean** | **-23.06** |
| | | | | **s.d.** | **0.08** |
| **Sucrose[b]** | 15-146-07 | 26-May-15 | OC | 20 | -12.08 |
| (n = 9) | 15-148-03 | 27-May-15 | OC | 20 | -12.40 |
| | 15-148-04 | 27-May-15 | OC | 20 | -12.31 |
| | | 5-Oct-17 | OC | 20 | -12.44 |
| | | 18-Apr-18 | OC | 20 | -12.04 |
| | | 18-Apr-18 | OC | 20 | -12.30 |
| | | 26-Feb-19 | OC | 20 | -12.21 |
| | | 26-Feb-19 | OC | 20 | -12.16 |
| | | 26-Feb-19 | OC | 20 | -12.04 |
| | | | | **mean** | **-12.22** |
| | | | | **s.d.** | **0.15** |
| **Rice Char** | 04-328-06 | 23-Nov-04 | OC | n/m | -24.42 |
| (n = 1) | 04-328-07 | 23-Nov-04 | POC | n/m | -26.67 |
| | 04-328-05 | 23-Nov-04 | EC | n/m | -26.94 |



| | | | | | | |
|---|---|---|---|---|---|---|
| | | fraction weighted | TC | 160 | **-26.74** | |
| **SRM-1649a** | | | | | | |
| (n = 2) | 04-330-03 | 25-Nov-04 | OC | n/m | -26.38 | |
| | 04-338-08 | 3-Dec-04 | OC | n/m | -26.29 | |
| | 04-330-05 | 25-Nov-04 | POC | n/m | -25.51 | |
| | 04-338-07 | 3-Dec-04 | POC | n/m | -25.66 | |
| | 04-330-06 | 25-Nov-04 | EC | n/m | -25.56 | |
| | 04-338-09 | 3-Dec-04 | EC | n/m | -25.43 | |
| | | fraction[c] weighted | TC | ~ 600 | **-25.84 ± 0.07** | |

[a]Sucrose was loaded as a solution (μg C), Regal Black, C1150, Rice char, and SRM-1649a as a powder (μg dry mass); [b]$\delta^{13}C_{VPDB}$ of bulk material (sucrose) via off-line method: -12.0 ± 0.2‰ (Satoshi, 2008); [c]Mean fraction (of two measurements) weighted isotopic composition of TC; n/m = not measured.



**Table S4.** Stable isotopic compositions of $^{13}C/^{12}C$ in OC and EC fractions from mixtures of reference materials. OC and EC fractions were isolated with the ECT9 protocol (Huang et al., 2006), purified in a vacuum system and analyzed on a MAT253 (Huang et al., 2013) at the CAIR lab, CRD, ASTD/ECCC.

| Reference mate | Lab ID | ate | Initial mass | | Measured fraction | $\delta^{13}C_{VPDB}$ |
|---|---|---|---|---|---|---|
| | | | Sucrose | Regal Black | | |
| | | | µg C | µg | | (‰) |
| **Regal Black** | 15-148-08 | 28-May-15 | 10 | 22 | EC | -27.49 |
| n = 9 | 15-148-05 | 28-May-15 | 15 | 26 | EC | -27.73 |
| | 15-149-07 | 29-May-15 | 20 | 50.4 | EC | -27.34 |
| | 15-148-09 | 28-May-15 | 30 | 66 | EC | -27.32 |
| | 16-224-04 | 11-Aug-16 | 20 | 57 | EC | -27.31 |
| | 16-224-07 | 11-Aug-16 | 20 | 53 | EC | -27.27 |
| | 16-224-08 | 11-Aug-16 | 20 | 58 | EC | -27.37 |
| | 16-225-07 | 12-Aug-16 | 10 | 20 | EC | -27.57 |
| | 17-248-08 | 30-Aug-17 | 20 | 53 | EC | -27.47 |
| | | | | | **mean** | **-27.43** |
| | | | | | **s.d.** | **0.15** |
| **Sucrose** | 15-149-04 | 29-May-15 | 10 | 22 | OC | -12.82 |
| n = 9 | 15-148-06 | 28-May-15 | 15 | 26 | OC | -12.54 |
| | 15-149-05 | 29-May-15 | 20 | 50.4 | OC | -12.54 |
| | 15-149-06 | 29-May-15 | 30 | 66 | OC | -12.29 |
| | 16-224-05 | 11-Aug-16 | 20 | 57 | OC | -13.04 |
| | 16-224-06 | 11-Aug-16 | 20 | 53 | OC | -12.36 |
| | 16-225-03 | 12-Aug-16 | 20 | 58 | OC | -12.72 |
| | 16-225-04 | 12-Aug-16 | 10 | 20 | OC | -12.86 |
| | 17-242-06 | 30-Aug-17 | 20 | 53 | OC | -12.34 |
| | | | | | **mean** | **-12.61** |
| | | | | | **s.d.** | **0.26** |

930





**Table S5.** Calculated stable isotopic composition ($^{13}C/^{12}C$) in a two-end-member-mixing system with endmember #1 being Sucrose ($\delta^{13}CV_{PDB}$ =-12.22‰) and end member #2 being Regal black ($\delta^{13}CV_{PDB}$=-27.61‰) and where endmember #1 is mixed into endmember#2.

| $\delta^{13}C_{VPDB}$ of pure endmember | | fraction of sucrose in mixture (Sucrose + Regal black) | $\delta^{13}C_{VPDB}$ of the mixture calculated |
|---|---|---|---|
| Sucrose | Regal black | | |
| ‰ | | % | ‰ |
| | | 0 | -27.610 |
| | | 1 | -27.456 |
| | | 2 | -27.302 |
| | | 3 | -27.148 |
| | | 4 | -26.994 |
| | | 5 | -26.841 |
| | | 10 | -26.071 |
| | | 20 | -24.532 |
| | | 30 | -22.993 |
| | | 40 | -21.454 |
| | | 50 | -19.915 |
| | | 60 | -18.376 |
| -12.22 | -27.61 | 70 | -16.837 |
| | | 80 | -15.298 |
| | | 90 | -13.759 |
| | | 91 | -13.605 |
| | | 92 | -13.451 |
| | | 93 | -13.297 |
| | | 94 | -13.143 |
| | | 95 | -12.990 |
| | | 96 | -12.836 |
| | | 97 | -12.682 |
| | | 98 | -12.528 |
| | | 99 | -12.374 |
| | | 100 | -12.220 |




**Table S6.** Radiocarbon content, expressed as fraction modern carbon (FM), of total (TC), organic (OC), and elemental (EC) carbon fractions with and without background correction following Santos et al. (2010). OC and EC fractions were isolated with the ECT9 protocol (Huang et al., 2006) from pure reference materials (into the form of $CO_2$), then purified cryogenically and sealed in ampoules at the CAIR lab, ECCC. $CO_2$ is reduced to graphite (Santo et al., 2007b, 2007a) and analyzed at the KCCAMS facility.

| UCIAMS# | Fraction | Mass after ECT9 µgC | Mass atKCCAMS µgC | Corrected FM | ± | Uncorrected FM | ± |
|---|---|---|---|---|---|---|---|
| **Adipicacid** | | | | | | | |
| 153279 | TC | 10 | 14 | -0.0050 | 0.0367 | 0.0593 | 0.0010 |
| 153280 | TC | 17 | 16 | -0.0116 | 0.0325 | 0.0465 | 0.0009 |
| 153281 | TC | 23 | 29 | -0.0043 | 0.0165 | 0.0268 | 0.0005 |
| 153282 | TC | 37 | 37 | -0.0102 | 0.0125 | 0.0140 | 0.0006 |
| mean | | | | -0.0078 | | | |
| s.d. | | | | 0.0037 | | | |
| **Sucrose** | | | | | | | |
| 153283 | TC | 5 | 7 | 1.0041 | 0.0885 | 0.8766 | 0.0101 |
| 153284 | TC | 5 | 7 | 1.0031 | 0.0878 | 0.8759 | 0.0051 |
| 153285 | TC | 5 | 7 | 1.0346 | 0.0938 | 0.8960 | 0.0064 |
| 153286 | TC | 10 | 11 | 1.0529 | 0.0516 | 0.9652 | 0.0045 |
| 153287 | TC | 10 | 11 | 1.0360 | 0.0511 | 0.9510 | 0.0070 |
| 153288 | TC | 10 | 12 | 1.0571 | 0.0510 | 0.9702 | 0.0056 |
| 153289 | TC | 20 | 21 | 1.0477 | 0.0265 | 1.0006 | 0.0069 |
| 153290 | TC | 20 | 21 | 1.0429 | 0.0257 | 0.9971 | 0.0058 |
| 153291 | TC | 20 | 21 | 1.0470 | 0.0262 | 1.0000 | 0.0056 |
| 153292 | TC | 40 | 41 | 1.0405 | 0.0127 | 1.0170 | 0.0034 |
| 153293 | TC | 40 | 38 | 1.0543 | 0.0139 | 1.0282 | 0.0034 |
| 153294 | TC | 40 | 42 | 1.0509 | 0.0125 | 1.0272 | 0.0026 |
| 153295 | OC | 20 | 20 | 1.0844 | 0.0290 | 1.0305 | 0.0041 |
| mean | | | | 1.0427 | | | |
| s.d. | | | | 0.0213 | | | |
| **C1150** | | | | | | | |
| 153303 | TC | 7 | 10 | 0.0310 | 0.0535 | 0.1154 | 0.0020 |
| 153304 | TC | 16 | 23 | 0.0278 | 0.0205 | 0.0644 | 0.0012 |
| 153305 | TC | 34 | 36 | -0.0012 | 0.0131 | 0.0237 | 0.0006 |
| 153306 | TC | 45 | 55 | 0.0041 | 0.0083 | 0.0201 | 0.0003 |





| 153307 | EC | 32 | 33 | -0.0072 | 0.0144 | 0.0202 | 0.0004 |
|---|---|---|---|---|---|---|---|
| mean | | | | 0.0109 | | | |
| s.d. | | | | 0.0174 | | | |
| **RegalBlack** | | | | | | | |
| 153308 | TC | 16 | 23 | 0.0161 | 0.0209 | 0.0540 | 0.0008 |
| 153309 | TC | 47 | 53 | -0.0008 | 0.0087 | 0.0160 | 0.0004 |
| 153310 | EC | 28 | 41 | -0.0057 | 0.0112 | 0.0159 | 0.0004 |
| mean | | | | 0.0032 | | | |
| s.d. | | | | 0.0114 | | | |
| **Ricechar** | | | | | | | |
| 153299 | TC | 6 | 7 | 0.9383 | 0.0830 | 0.8272 | 0.0097 |
| 153300 | TC | 12 | 15 | 1.0463 | 0.0390 | 0.9784 | 0.0057 |
| 153301 | TC | 24 | 22 | 1.0823 | 0.0254 | 1.0348 | 0.0046 |
| 153302 | EC | 13 | 15 | 1.0621 | 0.0383 | 0.9940 | 0.0046 |
| mean | | | | 1.0323 | | | |
| s.d. | | | | 0.0643 | | | |
| **OxalicacidII[a]** | | | | | | | |
| 153316 | TC | n/a | 7 | 1.3141 | 0.0398 | 1.2411 | 0.0203 |
| 153315 | TC | n/a | 17 | 1.3365 | 0.0137 | 1.3080 | 0.0063 |
| 153314 | TC | n/a | 45 | 1.3342 | 0.0051 | 1.3235 | 0.0027 |
| mean | | | | 1.3283 | | | |
| s.d. | | | | 0.0123 | | | |
| **Adipicacid[a]** | | | | | | | |
| 153318 | TC | n/a | 6 | -0.0020 | 0.0313 | 0.0544 | 0.0031 |
| 153317 | TC | n/a | 16 | -0.0016 | 0.0115 | 0.0205 | 0.0011 |
| 153278 | TC | n/a | 56 | -0.0014 | 0.0033 | 0.0051 | 0.0003 |
| mean | | | | -0.0017 | | | |
| s.d. | | | | 0.0003 | | | |

[a]Reference standards that underwent combustion and graphitization process only for blank determination at KCCAMS (without ECT9); n/a. = not applicable

940





**Table S7.** Radiocarbon content, expressed as fraction modern carbon (FM), of total (TC), organic (OC), and elemental (EC) carbon fractions with and
without background correction following Santos et al. (2010). OC and EC fractions were isolated with the ECT9 protocol (Huang et al., 2006) from
mixtures of reference materials (into the form of $CO_2$), then purified cryogenically and sealed in ampoules at ECCC.  $CO_2$ is reduced to graphite (Santos
al., 2007b, 2007a) and analyzed at KCCAMS facility.

| UCI AMS # | Fraction measured | Initial loaded mass | | Mass after ECT9 | Mass at KCCAMS | Corrected FM | | Uncorrected FM | |
|---|---|---|---|---|---|---|---|---|---|
| | | μg C | μg | μg C | | | ± | | ± |
| **Sucrose + Regal black** | Sucrose | Regal black | | | | | | | |
| 159800 | OC | 5 | 10 | 5 | 6 | 1.0568 | 0.0648 | 0.9738 | 0.0107 |
| 159802 | OC | 10 | 21 | 11 | 10 | 1.0542 | 0.0337 | 1.0057 | 0.0049 |
| 159804 | OC | 15 | 29 | 16 | 15 | 1.0629 | 0.0216 | 1.0298 | 0.0037 |
| 159806 | OC | 20 | 39 | 21 | 20 | 1.0436 | 0.0156 | 1.0201 | 0.0034 |
| 159808 | OC | 30 | 63 | 32 | 29 | 1.0563 | 0.0107 | 1.0395 | 0.0025 |
| 159801 | EC | 5 | 10 | 10 | 11 | -0.0361 | -0.0502 | 0.0535 | 0.0014 |
| 159803 | EC | 10 | 21 | 20 | 19 | -0.0189 | -0.0270 | 0.0317 | 0.0007 |
| 159805 | EC | 15 | 29 | 28 | 36 | -0.0091 | -0.0136 | 0.0172 | 0.0005 |
| 159807 | EC | 20 | 39 | 38 | 44 | 0.0014 | 0.0110 | 0.0226 | 0.0004 |
| 159809 | EC | 30 | 63 | 61 | 56 | 0.0019 | 0.0085 | 0.0186 | 0.0003 |
| **Adipic acid + Bulk rice char** | Adipic acid | Bulk rice char[a] | | | | | | | |
| 159822 | OC | 5 | 11 | 6 | 6 | 0.1009 | 0.0856 | 0.2279 | 0.0027 |
| 159824 | OC | 10 | 22 | 12 | 11 | 0.0759 | 0.0450 | 0.1516 | 0.0021 |
| 159826 | OC | 15 | 35 | 18 | 17 | 0.1078 | 0.0278 | 0.1558 | 0.0013 |
| 159828 | OC | 20 | 44 | 23 | 22 | 0.1072 | 0.0204 | 0.1432 | 0.0014 |
| 159830 | OC | 25 | 51 | 29 | 23 | 0.1552 | 0.0185 | 0.1868 | 0.0011 |
| 159832 | OC | 30 | 60 | 34 | 32 | 0.1013 | 0.0138 | 0.1263 | 0.0009 |
| 159823 | EC | 5 | 11 | 5 | 5 | 1.1063 | 0.0887 | 0.9903 | 0.0063 |
| 159825 | EC | 10 | 22 | 10 | 8 | 1.0981 | 0.0486 | 1.0263 | 0.0052 |
| 159827 | EC | 15 | 35 | 16 | 14 | 1.0559 | 0.0231 | 1.0211 | 0.0034 |
| 159829 | EC | 20 | 44 | 20 | 17 | 1.0619 | 0.0190 | 1.0328 | 0.0040 |
| 159831 | EC | 25 | 51 | 23 | 22 | 1.0625 | 0.0143 | 1.0400 | 0.0027 |
| 159833 | EC | 30 | 60 | 27 | 24 | 1.0633 | 0.0131 | 1.0426 | 0.0028 |
| **Adipic acid + Rice char_EC[b]** | Adipic acid | Rice char_EC | | | | | | | |
| 159810 | OC | 5 | 13 | 5 | 6 | -0.0605 | -0.1166 | 0.1212 | 0.0032 |
| 159812 | OC | 10 | 19 | 10 | 10 | -0.0324 | -0.0558 | 0.0655 | 0.0015 |



| 159814 | OC | 15 | 34 | 15 | 15 | -0.0075 | -0.0345 | 0.0556 | 0.0008 |
| 159816 | OC | 20 | 38 | 20 | 20 | 0.0107 | 0.0248 | 0.0568 | 0.0011 |
| 159818 | OC | 25 | 49 | 25 | 25 | -0.0009 | -0.0198 | 0.0366 | 0.0005 |
| 159820 | OC | 30 | 60 | 30 | 29 | 0.0103 | 0.0168 | 0.0421 | 0.0006 |
| 159811 | EC | 5 | 13 | 6 | 5 | 1.0926 | 0.0931 | 0.9755 | 0.0094 |
| 159813 | EC | 10 | 19 | 8 | 7 | 1.0702 | 0.0506 | 0.9997 | 0.0058 |
| 159815 | EC | 15 | 34 | 15 | 16 | 1.0709 | 0.0203 | 1.0392 | 0.0037 |
| 159817 | EC | 20 | 38 | 17 | 20 | 1.0726 | 0.0162 | 1.0471 | 0.0038 |
| 159819 | EC | 25 | 49 | 22 | 21 | 1.0749 | 0.0152 | 1.0505 | 0.0029 |
| 159821 | EC | 30 | 60 | 27 | 27 | 1.0723 | 0.0116 | 1.0535 | 0.0024 |

[a]The bulk rice char contains 52% of TC, on which 14% is OC and 86% EC, respectively; [b]Adipic acid was injected after the OC of rice char is removed through combustion at 870°C via ECT9. Thus, adipic acid was mixed only with rice char-EC, and the OC of the mixture is only from Adipic acid and EC of the mixture is only from Rice char.