# Peer review of "Application of the ECT9 protocol for radiocarbon-based source apportionment of carbonaceous aerosols"

_Atmospheric Measurement Techniques, 2020_

## Referee Comment (RC1) · Anonymous Referee #1 · 4 Sep 2020

This paper presents an evaluation of the ECT9 separation protocol for the measurement of radiocarbon in OC and EC. Radiocarbon measurements of OC and EC in carbonaceous aerosols provide a powerful tool for understanding the sources of these materials. Overall, the authors do an excellent job of describing the method, conducting the critical tests for validating the radiocarbon measurements, and comparing it to other standard methods currently in use. I recommend accepting the paper for publication with minor revisions noted below.

Instead of using the term "FM14C", I recommend using the term "F14C" as recommended by Reimer et al. (2004). Reimer, P. J., Brown, T. A., & Reimer, R. W. (2004).

[Figure]

Discussion: Reporting and calibration of post-bomb C-14 data. Radiocarbon, 46(3), 1299–1304. A better acronym for pyrolyzed organic carbon might be PyOC; the term POC means particulate organic carbon for researchers in the aquatic sciences.

Line 112. Rewrite "The fraction are separated based on their thermal refractory."

Section 2.3 There is a lot of detail in this section and some of it could be removed but osme of it enhanced. I am not clear as to how you load dissolved material onto a filter without losing some of it. Please explain.

Lines 210 on. There is a lot of reliance on the Santos et al. 2007 paper for assigning errors to the amount of extraneous carbon added during sample preparation. I'm sure more work has been done since then although perhaps not published. It might be useful to mention this unpublished work. However, an assignment of $\pm 50\%$ is very conservative and hard to argue with.

Line 219. Please better define what "14C analysis" refers to. Is it from graphite prep on or does is start later in the process.

Line 232. How was the mass determined at the CAIR lab? Is it from the integration of the OC/EC signals or from manometry? If it is from manometry, it is not a great comparison and probably does not warrant a figure.

Line 244 on. The data seem a bit iffy below 10 $\mu$g.

Line 260. I think it is optimistic to state that the technique is good for samples containing as little as 2 $\mu$g C. It definitely seems robust for samples containing >10 $\mu$g C and appears useful down to 5 $\mu$g C.

Lines 270-279. I am a little confused about the discussion of the rice char. If 14% of the carbon in the rice char is OC and rice char is modern, it would be expected that you would find modern carbon in the combined OC sample. The question is whether a mass balance indicates that the measured fraction modern is what one would expect.

[Figure]

Is it possible to redraw Figure 8 so that it is possible to see the peaks as robust features?

Lines 325 on. I find the comparison of the SRM 8785 analyses using the ECT9 and Swiss_4S protocols not as compelling as the previous figures. The results for the Swiss_4S protocol are difficult to interpret and more discussion is warranted. It certainly looks as though it would be very challenging to isolate OC from EC in the final peak in Figure 9C.

---

## Referee Comment (RC2) · Anonymous Referee #3 · 18 Nov 2020

1. Does the paper address relevant scientific questions within the scope of AMT? The manuscript fits perfectly within the scope of the journal Atmospheric Measurement Technique. It proposes the use of ECT9 protocol to separate OC and EC for radiocarbon analysis. This protocol is normally used for concentration and stable isotope measurements on OC and EC fractions, but this time the application is extended for 14C based source apportionment of carbonaceous aerosols. 2. Does the paper present novel concepts, ideas, tools, or data? The manuscript proposes the use of the ECT9 protocol to physically separate OC from EC in carbonaceous aerosols, but introduces new adaptations of the method to determine 14C of both fractions. The 13C composition of the fractions obtained with the method was also determined to as-

sure that the fractions were well separated. 3. Are substantial conclusions reached? Yes. The effectiveness of the ECT9 to physically separate OC and EC from aerosol samples for 13C and 14C analysis is demonstrated. 4. Are the scientific methods and assumptions valid and clearly outlined? Yes. 5. Are the results sufficient to support the interpretations and conclusions? Yes. 6. Is the description of experiments and calculations sufficiently complete and precise to allow their reproduction by fellow scientists (traceability of results)? Yes. The authors used a number of materials containing only OC or EC as well as mixtures of them. The experiments are clearly described. The tests to establish accuracy, precision and background are well described. HOWEVER, I would suggest some minor additions: Concerning the sample preparation protocol: In Section 2.3 Isolatin of OC, EC or TC... Line 150: Please give more details on how the filters are treated after an OC material dissolved in water is loaded onto a pre-cleaned quartz filter. Line 151: Please explain in a more detailed way how the filter punch is loaded and manipulated to avoid losing material during ECT9 protocol. Concerning the comparison of FM14C obtained values vs FM14C accepted values: Line 245: it is mentioned that FM14C values of pure modern and fossil reference materials agreed with their accepted FM14C values within aprox. 5% uncertainty. Please indicate the individual uncertainties that resulted in less than 5% in average. 7. Do the authors give proper credit to related work and clearly indicate their own new/original contribution? Yes. The cited references are adequate. 8. Does the title clearly reflect the contents of the paper? Yes. The title fully reflects the objective and the expected results 9. Does the abstract provide a concise and complete summary? Yes 10. Is the overall presentation well structured and clear? Yes 11. Is the language fluent and precise? The language is very appropriate. 12. Are mathematical formulae, symbols, abbreviations, and units correctly defined and used? Yes 13. Should any parts of the paper (text, formulae, figures, tables) be clarified, reduced, combined, or eliminated? The organization of the manuscript is adequate. The selected materials for testing the protocol are adequate; the description of methodology is in general clearly explained; except that a more detailed description on the filter loading. Validation of the protocol applied

to 14C analysis is rigorous. 14. Are the number and quality of references appropriate? Yes 15. Is the amount and quality of supplementary material appropriate? Yes.

---

## Author Response (AR1)

Thanks very much for the editor's effort to coordinate the reviewing and for the reviewer's constructive feedback and comments. We will answer the questions and address the concerns point by point raised by the reviewer#1 below in the format of "reviewer's comments/ author's responses".

This paper presents an evaluation of the ECT9 separation protocol for the measurement of radiocarbon in OC and EC. Radiocarbon measurements of OC and EC in carbonaceous aerosols provide a powerful tool for understanding the sources of these materials. Overall, the authors do an excellent job of describing the method, conducting the critical tests for validating the radiocarbon measurements, and comparing it to other standard methods currently in use. I recommend accepting the paper for publication with minor revisions noted below.
Appreciated for the general constructive feedback. Thanks for your effort.

Instead of using the term "FM14C", I recommend using the term "F14C" as recommended by Reimer et al. (2004). Reimer, P. J., Brown, T. A., & Reimer, R. W. (2004). 1299–1304. A better acronym for pyrolyzed organic carbon might be PyOC; the term POC means particulate organic carbon for researchers in the aquatic sciences.
We thank the reviewer for these suggestions!
"$F^{14}C$" has replaced "$FM^{14}C$" and PyOC has replaced POC through the entire text.

Line 112. Rewrite "The fraction are separated based on their thermal refractory."
The sentence is re-written as "The fractions are separated from each other according to their degree of refractory".

Section 2.3 There is a lot of detail in this section and some of it could be removed but some of it enhanced. I am not clear as to how you load dissolved material onto a filter without losing some of it. Please explain.
Only water soluble OC i.e., sucrose were injected onto filters. Please see the description at section 2.3 from L153-157 (in the revised version), which has been modified as
"*OC materials were first dissolved in MQ-water with known volumes to obtain their concentrations, and then a known amount of OC solution (5-10 µl) was very carefully applied onto a pre-cleaned quartz filter punch (1.5 cm², Pall Canada Limited) with a syringe. After the injection, the quartz boat holding the punch is pushed into the analyzer at the right position inside. The volume of OC solution used does not saturate the filter, but merely moistens the surface. After purging the filter for about 20 minutes ensuring the water is gone, the filter is ready for analysis*"
The loading of sucrose solution onto filters is a routine procedure of daily standard checks in OC/EC measurements. The liquid volume is so small that it moistens only the surface of the filter.

Lines 210 on. There is a lot of reliance on the Santos et al. 2007 paper for assigning errors to the amount of extraneous carbon added during sample preparation. I'm sure more work has been done since then although perhaps not published. It might be useful to mention this unpublished work. However, an assignment of ±50% is very conservative and hard to argue with.
Thanks to the reviewer for raising this point. As noted, although Santos et al., 2007 described modifications to the spectrometer, sample processing, mass balance correction approach and validation on how to effectively measure and report the results of small and ultra-small graphite target $^{14}C$, indeed, the

results in this work call for an error assignment more than our standard ± 30% of the blank. Santos et al., 2010 (cited in line 216) shows the long-term measurements of small to ultra-small blanks of combustible reference materials as well as the blanks related to processing chemicals. In the work by Santos et al., 2010 and many subsequent ones, reporting blank assessment for specific case studies (e.g., Fernandez et al. 2014, Mouteva et al. 2015, Reyerson et al. 2016), we confirmed that a more conservative assignment to error propagation into individual uncertainties would be suitable for a method evaluation. Basically, long-term evaluations of the blank are essential to determine its variance. Therefore, we prefer to maintain a large assignment of blank error for the ECT9 method for now. A possible reduction of it will be pursued in future works.

Line 219. Please better define what "14C analysis" refers to. Is it from graphite prep on or does is start later in the process.
The reviewer raised a good point. The "$^{14}$C analysis" in the manuscript referred to "$^{14}$C sample preparation and analysis"

Line 232. How was the mass determined at the CAIR lab? Is it from the integration of the OC/EC signals or from manometry? If it is from manometry, it is not a great comparison and probably does not warrant a figure.
The mass values in Figure 2- a. by the CAIR lab were determined by gravimetric methods either via weighing on a 6 or 7 digit balance for individual references, including Regal black, C1150, Rice char, Adipic acid or volumetric injection for sucrose solution. Whereas the mass values in Figure 2-b) were determined by the combination of gravimetric methods mentioned with OC/EC ratios in Table 1 (i.e., adipic + Rice char, regal black + sucrose). As the methods used at CAIR lab are different from the manometric method used at KCCAMS lab, the comparison does show the importance of the final mass recovery at KCCAMS for those were originally loaded via gravimetric or volumetric methods through ECT9 protocol.

Line 244 on. The data seem a bit iffy below 10 μg
On average, results for all individual (Table S6) and mixed reference materials (Table S7) are within 2±3% of their corresponding consensus values (Table 2). For samples containing > 10 μg C the results are within 1±1%, whereas samples containing between > 5 μg C and < 10 μg C they are around 7±5% in average, respectively. We modified the content as
*"The $F^{14}C$ values of the pure modern or fossil reference materials generally agreed with their accepted $F^{14}C$ values for both OC and EC fractions (within approximately 5% uncertainty on average, Fig. 3 and Table 2, Tables S6 & S7) after applying a constant amount $C_{ex}$ correction in $F^{14}C$ determination. Specifically, the overall agreements for all individual pure (Table S6) and mixed reference materials (Table S7, excluding the OC data from adipic acid + bulk rice char) are within 2±3% of their corresponding values (Table 2). On average, for samples containing >10μg C the agreements are within 1±1%, whereas samples containing between > 5 μg C and < 10 μg C they are around 7±5%, respectively."*

Line 260. I think it is optimistic to state that the technique is good for samples containing as little as 2 μg C. It definitely seems robust for samples containing >10 μg C and appears useful down to 5 μg C.
We agree that we may have overestimated the lower limit of the method here. Initially, we reported the minimal sample size based on the ECT9 blank size only. We have made some changes in response to this comments and are now reporting the minimum sample size for unknown samples base on what we confidently measured so far (i.e., approximately 5 μg C as suggested by the reviewer). We have modified the content as mentioned above.

Lines 270-279. I am a little confused about the discussion of the rice char. If 14% of the carbon in the rice char is OC and rice char is modern, it would be expected that you would find modern carbon in the combined OC sample. The question is whether a mass balance indicates that the measured fraction modern is what one would expect.

The assessment of the reviewer is correct. The initially unremoved ~14% OC fraction from the modern rice char was found in the isolated fossil adipic acid (pure fossil OC) after running the ECT9 method. Consequently, final $^{14}$C results of OC from the sequence UCIAMS#s 159822-159832 (Table S7) are elevated, rather than closer to zero. The results yielded an average F$^{14}$C value of $0.1081 \pm 0.0259$ (n=6) after blank corrections. Based on this value, a mass balance calculation indicates that $11 \pm 3\%$ of OC-Rice char is present. This estimate is close to the lower limit within the validity range to what one would expect. We have expanded our discussion of the effects of the unremoved 14%-OC fraction from the modern rice char mentioned by the reviewer, and added a couple of statements in this section to clarify this point.

Is it possible to redraw Figure 8 so that it is possible to see the peaks as robust features?
Yes, it is possible to re-draw Figure 8 to change the scale to show peak features more clearly (although the extent of change is limited due to a consistency with other figures). Figure 8 has been redrew.

[Figure]

Lines 325 on. I find the comparison of the SRM 8785 analyses using the ECT9 and Swiss_4S protocols not as compelling as the previous figures. The results for the Swiss_4S protocol are difficult to interpret and more discussion is warranted. It certainly looks as though it would be very challenging to isolate OC from EC in the final peak in Figure 9C.

Thanks to the reviewer for raising this concern. We modified the content as

*" It is important to note that to obtain EC fraction, the Swiss-4 (Table 3) method calls for filter sample pre-treatment, i.e., extraction with water before the thermal separation of OC/EC to minimize the contribution of charred OC from the 3ʳᵈ stage to EC at the 4ᵗʰ stage (Zhang et al., 2012). However, for a method comparison, the thermogram shown in Fig. 9c was from a filter without pre-treatment. While it is difficult to make direct comparisons between OC and EC from b) and c) in Figure 9, the laser profiles of those thermograms in Fig. a) and b) indicate that in both cases charred OC is negligible or minimum via ECT9."*

**Response to Interactive comment on "Application of the ECT9 protocol for radiocarbon-based source apportionment of carbonaceous aerosols"**
By Anonymous **Reviewer #2** ()

We thank the editor for coordinating the review and the reviewer for the constructive feedback.  Here, we answer the questions and address the concerns raised by the reviewer#2 point by point in the format of "reviewer's comments/ author's responses".

1. Does the paper address relevant scientific questions within the scope of AMT?
   The manuscript fits perfectly within the scope of the journal Atmospheric Measurement Technique.  It proposes the use of ECT9 protocol to separate OC and EC for radiocarbon analysis. Tis protocol is normally used for concentration and stable isotope measurement on OC and EC fractions, but this time the application is extended for $^{14}$C based source apportionment of carbonaceous aerosols.
2. Does the paper present novel concepts, ideas, tools, or data?
   The manuscript proposes the use of the ECT9 protocol to physically separate OC from EC in carbonaceous aerosols. The $^{13}$C composition of the fractions obtained with the method was also determined to assure that the fractions were well separated.
3. Are substantial conclusions reached?
   Yes, the effectiveness of the ECT9 to physically separate OC and EC from aerosol samples for $^{13}$C and $^{14}$C analysis is demonstrated.
4. Are the scientific methods and assumptions valid and clearly outlined?
   Yes
5. Are the results sufficient to support the interpretations and conclusions?
   Yes.
6. Is the description of experiments and calculations sufficiently complete and precise to allow their reproduction by fellow scientists (traceability of results)?
   Yes, the authors used a number of materials containing only OC or EC as well as mixtures of them.  The experiments are clearly described. The tests to establish accuracy, precision and background are well described.
   We appreciate the reviewer's support of our work

   However, I would suggest some minor additions:
   Concerning the sample preparation protocol:
   In section 2.3 isolating of OC, EC or TC

   Line 150: Please give more details on how the filter are treated after an OC material dissolved in water is loaded onto a pre-cleaned quartz filter.
   Reviewer #1 also raised this issue and we have amended the text in the revised version according to these details below:

*"Filters before mass loading were pre-combusted at 900 °C in a muffle furnace overnight and wrapped into pre-fired aluminum foil before cooling below 200 °C. Usually, OC materials were first dissolved in MQ-water with known volume to obtain its concentration, and then a known amount (5-10 µl) of OC solution was very carefully applied onto a pre-cleaned quartz filter surface (1.5 cm², Pall Canada Limited) via a syringe injection. After the injection, the quartz boat holding the punch is pushed to the right position inside of the analyzer. The volume of OC solution used does not saturate the filter, but merely moistens the surface. After purging the filter for about 20 minutes ensuring the water is gone, the filter is ready for analysis. EC (i.e., Regal black and C1150) and mixed materials (rice char or SRM 1649a), which cannot be completely dissolved in water, were directly weighed onto pre-cleaned quartz filter punches in form of solids (powders). Adipic acid were also loaded as powder. The final power mass was determined by the difference weighted before and after analysis. A filter punch with the loaded mass was carefully carried to the Sunset analyzer by a Pyrex glass Petri dish with cover for analysis with the ECT9 protocol."*

Line 151: Please explain in a more detailed way how the filter punch is loaded and manipulated to avoid losing material during ECT9 protocol.

Soluble (OC) materials are loaded onto filters as described above. Insoluble (EC) materials are loaded in form of solids (powders). A punch of pre-cleaned quartz filter is loaded with powder and weighed directly, via a 6 or 7 digit balance, before and after analysis. The mass of the powder is calculated by difference. The quartz punch with the power is carefully carried within a glass petri dish with cover to the analyzer for analysis via ECT9 protocol.
We modified the content between L151 -160 in the original version to include the details of mass loading for dissolved OC and solid powers.

Concerning the comparison of $FM^{14}C$ obtained values vs. $FM^{14}C$ accepted values:
Line 245: It is mentioned that $FM^{14}C$ values of pure modern and fossil reference materials agreed with their accepted $FM^{14}C$ values within approx. 5% uncertainty. Please indicate the individual uncertainties that resulted in less than 5% in average.
We thank the reviewer for the careful examination of our work!

The 5% uncertainty mentioned in the paper is based on that
- the overall agreements of all individual pure (Table S6) and mixed reference materials (Table S7, excluding the OC data from adipic acid + bulk rice char) are within 2±3% of their corresponding consensus values (Table 2).
- For samples containing > 10 µg C the agreements are within 1±1%,
- whereas samples containing between > 5 µg C and < 10 µg C they are around 7±5% in average, respectively.
The mean values of $F^{14}C$ for individual mixed OC and EC measurements have been added in Table S7. We have included the following content in the revised version.

*" The $F^{14}C$ values of the pure modern or fossil reference materials generally agreed with their accepted $F^{14}C$ values for both OC and EC fractions (**within approximately 5% uncertainty on average, Fig. 3 and Table 2, Tables S6, S7**) after applying a constant amount $C_{ex}$ correction in $F^{14}C$ determination. Specifically, the overall agreements for all individual pure (Table S6) and mixed reference materials (Table S7) are within 2±3% of their corresponding consensus value (Table 2). On average, for samples containing > 10 µg C the agreements are within 1±1%, whereas samples containing between > 5 µg C and < 10 µg C they are around 7±5%, respectively."*

7. Do the authors give proper credit to related work and clearly indicate their own new/original contribution?
   Yes, the cited references are adequate.
8. Does the title clearly reflect the contents of the paper?
   Yes, the title fully reflects the objective and the expected results.
9. Does the abstract provide a concise and complete summary?
   Yes.
10. Is the overall presentation well-structured and clear?
    Yes.
11. Is the language fluent and precise?
    The language is very appropriate.
12. Are mathematical formulae, symbols, abbreviations, and units correctly defined and used?
    Yes.
13. Should any parts of the paper (text, formulae, figures, tables) be clarified, reduced, combined, or eliminated?
    The organization of the manuscript is adequate. The selected materials for testing the protocol are adequate; the description of methodology is in general clearly explained; except that a more detailed description on the filter loading. Validation of the protocol applied to $^{14}C$ analysis is rigorous.
14. Are the number and quality of references appropriate?
    Yes.
15. Is the amount and quality of supplementary material appropriate?
    Yes.

Thank you for the constructive comments.